# PWM: Policy Learning with Multi-Task World Models

**Ignat Georgiev** [1], **Varun Giridhar** [1], **Nicklas Hansen** [2], **Animesh Garg** [1]

[1] Georgia Institute of Technology     [2] UC San Diego

## Abstract

Reinforcement Learning (RL) has made significant strides in complex tasks but struggles in multi-task settings with different embodiments. World model methods offer scalability by learning a simulation of the environment but often rely on inefficient gradient-free optimization methods for policy extraction. In contrast, gradient-based methods exhibit lower variance but fail to handle discontinuities. Our work reveals that well-regularized world models can generate smoother optimization landscapes than the actual dynamics, facilitating more effective first-order optimization. We introduce Policy learning with multi-task World Models (PWM), a novel model-based RL algorithm for continuous control. Initially, the world model is pre-trained on offline data, and then policies are extracted from it using first-order optimization in less than 10 minutes per task. PWM effectively solves tasks with up to 152 action dimensions and outperforms methods that use ground-truth dynamics. Additionally, PWM scales to an 80-task setting, achieving up to 27% higher rewards than existing baselines without relying on costly online planning. Visualizations and code are available at `imgeorgiev.com/pwm`.

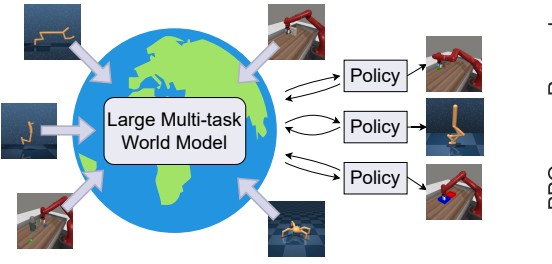
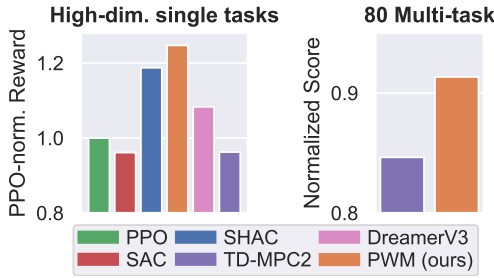

**Figure 1:** We propose PWM, a new method for multi-task RL that utilizes pre-trained world models to learn policies for each task. When sufficiently regularized, these world models induce smooth optimization landscapes, which allows for efficient first-order optimization. Our approach can solve tasks in <10 minutes and achieves higher rewards in both single-task and multi-task environments.

## 1 Introduction

The pursuit of generalizability in machine learning has recently been propelled by the training of large models on substantial datasets Brown et al. (2020); Kirillov et al. (2023); Bommasani et al. (2021). Such advancements have notably permeated robotics, where multi-task behavior cloning techniques have shown remarkable performance Zitkovich et al. (2023); Octo Model Team et al. (2024); Goyal et al. (2023); Bousmalis et al. (2023). Nevertheless, these approaches predominantly hinge on near-expert data and struggle with adaptability across diverse robot morphologies due to their dependence on teleoperation Zitkovich et al. (2023); Octo Model Team et al. (2024); Kumar et al. (2021).

In contrast, Reinforcement Learning (RL) offers a robust framework capable of learning from suboptimal data, addressing the aforementioned limitations. However, traditional RL has been focused on single-task experts Mnih et al. (2013); Schulman et al. (2017); Haarnoja et al. (2018). Recently, Hansen et al. (2024) suggested that a potential pathway to multi-task RL is with the world

models framework, where a large model learns the environment dynamics and is then combined with Zeroth-order Gradient (ZoG) methods. Despite advancements, ZoG methods struggle with sample inefficiency due to the high variance Mohamed et al. (2020); Suh et al. (2022); Parmas et al. (2023) and online planning time scales with model size, rendering it infeasible at scale.

First-order Gradient (FoG) methods provide a low-variance alternative that have shown superior sample efficiency and asymptotic performance when combined with smooth differentiable simulations Xu et al. (2022). However, they struggle to optimize through discontinuities Suh et al. (2022); Georgiev et al. (2024). In this work, we explore the tight coupling between FoG optimization and world models through the lens of differentiable simulation. Counter-intuitively, we find that for gradient-based optimization, we don't want world models to be accurate; instead, we want them to be smooth and have a low optimality gap. This in turn enables efficient FoG optimization.

Building on these insights, we propose Policy learning with multi-task World Models (PWM), an algorithm that can learn policies from offline pre-trained world models in under <10 minutes per task. With this new-found efficiency, we also propose a *new multi-task framework*, where instead of training a multi-task algorithm, we only train a multi-task world model and then extract a policy for each task. This decoupling of the supervised and RL objectives results in more stable and efficient learning with higher episode rewards. Our empirical evaluations on high-dim. tasks indicate that PWM not only achieves higher reward than baselines but also outperforms methods that use ground-truth dynamics. In a multi-task scenario utilizing a pre-trained 48M parameter world model from TD-MPC2, PWM achieves up to 27% higher reward than TD-MPC2 without relying on online planning. This underscores the efficacy of PWM and supports our broader contributions:

1. **Correlation Between World Model Smoothness and Policy Performance:** Through pedagogical examples and ablations, we demonstrate that smoother, better-regularized world models significantly enhance policy performance. Notably, this results in an inverse correlation between model accuracy and policy performance.

2. **Efficiency of First-Order Gradient (FoG) Optimization:** We show that combining FoG optimization with well-regularized world models enables more efficient policy learning compared to zeroth-order methods. Furthermore, policies learned from world models asymptotically outperform those trained with ground-truth simulation dynamics, emphasizing the importance of the tight relationship between FoG optimization and world model design.

3. **Scalable Multi-Task Algorithm:** Instead of training a single multi-task policy model, we propose PWM, a framework where a multi-task world model is first pre-trained on offline data. Then per-task expert policies are extracted in <10 minutes per task, offering a clear and scalable alternative to existing methods focused on unified multi-task models.

## 2 BACKGROUND

We focus on discrete-time and infinite-horizon Reinforcement Learning (RL) scenarios characterized by system states $s \in \mathbb{R}^n = \mathcal{S}$, actions $a \in \mathbb{R}^m = \mathcal{A}$, dynamics function $f : \mathcal{S} \times \mathcal{A} \to \mathcal{S}$ and a reward function $r : \mathcal{S} \times \mathcal{A} \to \mathbb{R}$. Combined, these form a Markov Decision Problem (MDP) summarized by the tuple $(\mathcal{S}, \mathcal{A}, f, r, \gamma)$ where $\gamma$ is the discount factor. Actions at each timestep $t$ are sampled from a stochastic policy $a_t \sim \pi_\theta(\cdot|s_t)$, parameterized by $\theta$. The goal of the policy is to maximize the cumulative discounted rewards:

$$\max_{\theta} J(\theta) := \max_{\theta} \mathbb{E}_{\substack{s_1 \sim \rho(\cdot) \\ a_t \sim \pi_\theta(\cdot|s_t)}} \left[ \sum_{t=1}^{\infty} \gamma^t r(s_t, a_t) \right] \tag{1}$$

where $\rho(s_1)$ is the initial state distribution. Since this maximization over an infinite sum is intractable, in practice we often maximize over a value estimate. The value of a state $s_t$ is defined as the expected reward following the policy $\pi_\theta$

$$V_\psi^\pi(s_t) := \mathbb{E}_{a_h \sim \pi_\theta(\cdot|s_h)} \left[ \sum_{h=t}^{\infty} \gamma^h r(s_h, a_h) \right] \tag{2}$$

When $V$ is approximated with a learned model with parameters $\psi$ and $\pi_\theta$ attempts to maximize some function of $V$, we arrive at the popular and successful actor-critic architecture Konda & Tsitsiklis (1999). Additionally, in MBRL it is common to also learn approximations of $f$ and $r$, which we denote as $F_\phi$ and $R_\phi$, respectively. It has also been shown to be beneficial to encode the true state

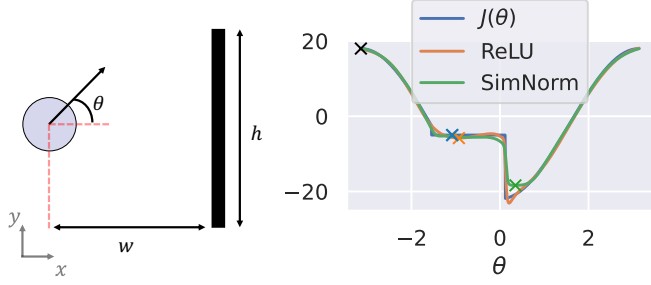

**(a)** Ball-wall visualization.  **(b)** Problem landscape.  **(c)** Model error and optimality gap.

| Model | Model error | Opt. gap |
|---|---|---|
| True | 0.0 | 16.85 |
| ReLU | 0.71 | 16.05 |
| SimNorm | 1.13 | 3.47 |

**Figure 2: Ball-wall pedagogical example.** The **left** figure visualizes the problem. The middle figure shows the problem landscape induced by each model. $J(\theta)$ shows the true underlying function and the two other are MLPs with different activation functions. We minimize each of these problems using gradient descent and starting at $\theta = -\pi$ (marker $\times$). The colored crosses represent the solutions converged to for each model. The **right** table shows the model approximation error during training and the optimality gap $|J(\theta^*) - J(\hat{\theta})|$ between the global minimum $\theta^*$ and the solution found for each model $\hat{\theta}$.

$s$ into a latent state $z$ using a learned encoder $E_\phi$ Hafner et al. (2019); Hansen et al. (2022; 2024); Hafner et al. (2023). Putting together all of these components, we can define a model-based actor-critic algorithm to consist of the tuple $(\pi_\theta, V_\psi, E_\phi, F_\phi, R_\phi)$ which can describe popular approaches such as Dreamer Hafner et al. (2019; 2023) and TD-MPC2 Hansen et al. (2024). Notably, we make an important distinction between the types of components. We refer to $E_\phi$, $F_\phi$ and $R_\phi$ as the *world model components* since they are supervised learning problems with fixed targets. On the other hand, $\pi_\theta$ and $V_\psi$ optimize for moving targets, which is fundamentally more challenging, and we refer to them as the *policy components*.

## 3 POLICY OPTIMIZATION THROUGH WORLD MODELS

This paper builds on the insight that since access to $F_\phi$ and $R_\phi$ is assumed through a pre-trained world model, we have the option to optimize Eq. 1 via *First-order Gradient (FoG) optimization* which exhibit lower gradient variance, more optimal solutions, and improved sample efficiency Mohamed et al. (2020). In our setting, these types of gradients are obtained by directly differentiating the expected terms of Eq. 1 as shown in Eq. 3. Note that this gradient estimator is also known as reparameterized gradient Kingma et al. (2015) and pathwise derivative Schulman et al. (2015). While we use the explicit $\nabla^{[1]}$ notation below, we later drop it for simplicity as all gradient types in this work are first-order gradients.

$$\nabla_{\boldsymbol{\theta}}^{[1]} J(\boldsymbol{\theta}) := \mathbb{E}_{\substack{\boldsymbol{s}_1 \sim \rho(\cdot) \\ \boldsymbol{a}_h \sim \pi_{\boldsymbol{\theta}}(\cdot|\boldsymbol{s}_h)}} \left[ \nabla_{\boldsymbol{\theta}} \left( \sum_{t=1}^{\infty} \gamma^t r(\boldsymbol{s}_t, \boldsymbol{a}_t) \right) \right] \tag{3}$$

As $\nabla_{\boldsymbol{\theta}}^{[1]} J(\boldsymbol{\theta})$ in itself is a random variable, we need to estimate it. A popular way to do that in practice is via Monte-Carlo approximation, where we are interested in two properties: bias and variance. In Sections 3.1 and 3.2 we tackle each aspect with a toy robotic control problem to build intuition. In Section 3.3, we combine our findings to propose a new algorithm.

### 3.1 LEARNING THROUGH CONTACT

FoGs are unbiased $\mathbb{E}\left[\bar{\nabla}^{[1]} J(\boldsymbol{\theta})\right] := \mathbb{E}\left[\sum_{n=1}^{N} \hat{\nabla}_n^{[1]} J(\boldsymbol{\theta})\right] = \nabla J(\boldsymbol{\theta})$, only if both the dynamics $f$ and rewards $r$ are Lipschitz-smooth Suh et al. (2022). However, many robotic problems involving contact are inherently non-smooth, which breaks these conditions and results in gradient sample error where $\mathbb{E}\left[\bar{\nabla}^{[1]} J(\boldsymbol{\theta})\right] \neq \nabla J(\boldsymbol{\theta})$ under a finite number of samples $N$. Instead of directly optimizing the true, discontinuous objective, it is advantageous to optimize a smooth surrogate, such as a model learned by a regularized deep neural network.

To illustrate this concept, we use a toy problem where a ball is thrown toward a wall at a fixed velocity, as shown in Figure 2a. The objective is to find the optimal initial angle $\theta$ such that we maximize

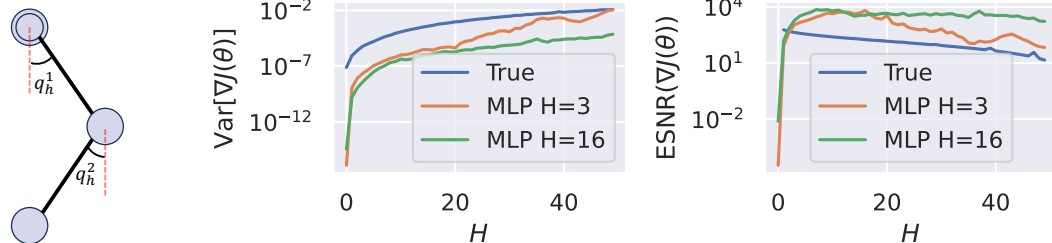

**Figure 3: Double pendulum pedagogical example.** The middle figure evaluates the variance of policy gradient estimates over $N = 100$ Monte-Carlo samples for varying horizons $H$. The right figure shows the same data but plots the Expected Signal-to-Noise ratio (ESNR) with higher values translating to more useful gradients. These results suggests that world models trained over long horizon trajectories provide more useful gradients. Note that $H = 3$ and $H = 16$ in the figure legends refer to the training horizon of the models.

forward distance. In this simplified pedagogical example, we assume that the ball "sticks" to the wall, creating a discontinuous optimization landscape (Figure 2b). We compare the performance of two models in approximating this objective: a 2-layer Multi-Layer Perceptron (MLP) with ReLU activation and another MLP with SimNorm activation Hansen et al. (2024) in the intermediate layers. SimNorm normalizes a latent vector $z$ by projecting it into simplices with dimension $P$ using a softmax operator. Given an input vector $z$, SimNorm can be expressed as a mapping into $L$ vectors:

$$\text{SimNorm}(z) := [g_1, ..., g_L], \quad g_i = \text{Softmax}(z_{i:i+P}) \tag{4}$$

We trained both MLPs and observed their effects on smoothing the optimization landscape (Figure 2b). The ReLU-activated MLP smooths the landscape but introduces a local minimum that hinders gradient descent, particularly when starting from $\theta = -\pi$, resulting in a large optimality gap (difference between the solution and the optimal solution: $\|\hat{\theta} - \theta^*\|$). In contrast, the SimNorm-activated MLP has additional regularization, which reduces the optimality gap at the expense of model accuracy (Table 2c). This example highlights that more accurate models do not always lead to better policies, as noted by Lambert et al. (2020). Our findings extend this by showing that for FoG optimization, prioritizing smoothness over accuracy can lead to improved results. Further details are provided in Appendix A.

## 3.2 LEARNING WITH CHAOTIC DYNAMICS

While FoGs have lower variance per step, they can accumulate significant variance over long-horizon rollouts Metz et al. (2021). Suh et al. (2022) link this variance to the smoothness of models and the length of the prediction horizon: $\text{Var}\left[\nabla J^{[1]}\right] \propto \|\nabla f(s, a)\|^{2H}$. At sufficiently high $H$, the high variance renders FoGs ineffective in chaotic systems. Chaotic systems are characterized by their sensitivity to initial conditions, where small perturbations can lead to exponentially divergent trajectories, making long-term prediction particularly challenging. The double pendulum, also known as the Acrobot Murray & Hauser (1991), is a classic example of such a system (Figure 3).

We analyze the variance of gradient estimators in the double pendulum using both the true dynamics and a SimNorm-activated MLP model. The MLP model was trained for auto-regressive prediction horizons of $H = 3$ and $H = 16$ until convergence on a large dataset. Figure 3 shows that both learned models exhibit reduced variance compared to the true dynamics. However, as noted by Parmas et al. (2023), variance alone is insufficient for drawing definitive conclusions about gradient quality. Instead, they propose analyzing gradients via their Expected Signal-to-Noise Ratio (ESNR), defined as:

$$\text{ESNR}(\nabla J(\boldsymbol{\theta})) = \mathbb{E}\left[\frac{\sum \mathbb{E}\left[\nabla^{[1]} J(\boldsymbol{\theta})\right]^2}{\sum \text{Var}\left[\nabla^{[1]} J(\boldsymbol{\theta})\right]}\right] \tag{5}$$

In Figure 3, we observe that learned models exhibit higher ESNR than the true dynamics, providing more useful gradients. Notably, the training horizon plays a critical role, with the $H = 16$ model sustaining a higher ESNR over higher $H$. We conclude that learned world models offer more informative policy gradients than the true system dynamics. Further details in Appendix B.

### 3.3 PWM: POLICY LEARNING WITH MULTI-TASK WORLD MODELS

Given the results from the previous subsection, we propose to view world models not as components of RL methods but instead as scalable, differentiable physics simulators that provide gradients with low sample error and variance. It is worth noting that approaches such as TD-MPC2 Hansen et al. (2024) do not exploit these properties but rather choose to optimize policies via DDPG-style gradients:

$$\nabla_{\boldsymbol{\theta}} J(\boldsymbol{\theta}) \approx \mathbb{E}_{\boldsymbol{a} \sim \pi(\cdot|s)}[\nabla_{\boldsymbol{\theta}} Q(\boldsymbol{s}, \boldsymbol{a})].$$

We propose a new method and framework for efficiently learning policies from large multi-task world models.

**Framework.** Assuming availability of data from multiple tasks, we first train a multi-task world model to predict future states and rewards. Then for each task we want to solve, we learn a single policy in minutes using FoG optimization. The policy is then deployed to solve the task and optionally fine-tune its world model and policy.

**Method.** For policy learning, we propose on-policy actor-critic approach inspired by differentiable simulation approaches Xu et al. (2022) where the actor is trained via FoG back-propagated through the world model, while the

---

**Algorithm 1:** PWM: Policy learning with multi-task World Models

**Given**: Multi-task dataset $\mathcal{B}$
**Given**: $\gamma$: discount rate
**Given**: $\alpha_{\boldsymbol{\theta}}, \alpha_{\boldsymbol{\psi}}, \alpha_{\boldsymbol{\phi}}$: learning rates
**Initialize learnable parameters** $\boldsymbol{\theta}, \boldsymbol{\psi}, \boldsymbol{\phi}$
▷ Pre-train world model once
**for** *N epochs* **do**
   $\boldsymbol{s}_{1:H}, \boldsymbol{a}_{1:H}, r_{1:H}, \boldsymbol{e} \sim \mathcal{B}$
   $\phi \leftarrow \phi - \alpha_{\boldsymbol{\phi}} \nabla \mathcal{L}_{wm}(\phi)$   ▷ Eq. 10
**end**
▷ Train policy on task embedding $\boldsymbol{e}$
**for** *M epochs* **do**
   $\boldsymbol{s}_1 \sim \mathcal{B}$
   $\boldsymbol{z}_1 = E_{\phi}(\boldsymbol{s}_1, \boldsymbol{e})$
   **for** *h=[1, ..., H]* **do**   ▷ Rollout
      $\boldsymbol{a}_h \sim \pi_{\boldsymbol{\theta}}(\cdot|\boldsymbol{z}_h)$
      $r_h = R_{\phi}(\boldsymbol{z}_h, \boldsymbol{a}_h, \boldsymbol{e})$
      $\boldsymbol{z}_{h+1} = F_{\phi}(\boldsymbol{z}_h, \boldsymbol{a}_h, \boldsymbol{e})$
   **end**
   $\boldsymbol{\theta} \leftarrow \boldsymbol{\theta} + \alpha_{\boldsymbol{\theta}} \nabla \mathcal{L}_{\pi}(\boldsymbol{\theta})$   ▷ Eq. 6
   $\boldsymbol{\psi} \leftarrow \boldsymbol{\psi} - \alpha_{\boldsymbol{\psi}} \nabla \mathcal{L}_V(\boldsymbol{\psi})$   ▷ Eq. 7-9
**end**

---

critic is trained via TD($\lambda$). The key to our approach is that training is done in a batched fashion where multiple trajectories are imagined in parallel. The actor loss function is akin to Eq. 1 but features rewards over a fixed horizon $H$, terminal value bootstrapping and usage of the learned world model components:

$$\mathcal{L}_{\pi}(\boldsymbol{\theta}) := \mathbb{E}_{\substack{\boldsymbol{s}_1 \sim \rho(\cdot) \\ \boldsymbol{a}_h \sim \pi_{\boldsymbol{\theta}}(\cdot|\boldsymbol{z}_h)}} \left[ \sum_{h=1}^{H-1} \gamma^h R_{\phi}(\boldsymbol{z}_h, \boldsymbol{a}_h) + \gamma^H V_{\boldsymbol{\psi}}(\boldsymbol{z}_H) \right] \quad \text{where} \quad \substack{\boldsymbol{z}_1 = E_{\phi}(\boldsymbol{s}_1) \\ \boldsymbol{z}_{t+1} = F_{\phi}(\boldsymbol{z}_t, \boldsymbol{a}_t)} \quad (6)$$

The critic is trained in a model-free fashion using TD($\lambda$) over an $H$-step rollout in latent space $\boldsymbol{z}$ as seen in other similar on-policy methods Sutton & Barto (2018); Hafner et al. (2019); Xu et al. (2022):

$$V_h(\boldsymbol{z}_t) := \sum_{n=t}^{t+h-1} \gamma^{n-t} R_{\phi}(\boldsymbol{z}_n, \boldsymbol{a}_n) + \gamma^{t+h} V_{\boldsymbol{\psi}}(\boldsymbol{z}_{t+h}) \quad (7)$$

$$\hat{V}(\boldsymbol{z}_t) := (1-\lambda) \left[ \sum_{h=1}^{H-t-1} \lambda^{h-1} V_h(\boldsymbol{z}_t) \right] + \lambda^{H-t-1} V_H(\boldsymbol{z}_t) \quad (8)$$

$$\mathcal{L}_V(\boldsymbol{\psi}) := \sum_{h=t}^{t+H} \left\| V_{\boldsymbol{\psi}}(\boldsymbol{z}_h) - \hat{V}(\boldsymbol{z}_h) \right\|_2^2 \quad (9)$$

We use an ensemble of 3 critics to reduce variance. To enable FoG optimization, it is important to use a well-regularized world model. We use the $\left( E_{\phi}(\boldsymbol{s}, \boldsymbol{e}), F_{\phi}(\boldsymbol{s}, \boldsymbol{a}, \boldsymbol{e}), R_{\phi}(\boldsymbol{s}, \boldsymbol{a}, \boldsymbol{e}) \right)$ model proposed by TD-MPC2 Hansen et al. (2024) with learnable task embeddings $\boldsymbol{e}$. It is trained in an auto-regressive fashion by sampling data from a buffer with loss function:

$$\mathcal{L}_{\text{wm}}(\phi) = \mathbb{E}_{(\boldsymbol{s}, \boldsymbol{a}, r, \boldsymbol{s}', \boldsymbol{e})_{0:H} \sim \mathcal{B}} \left[ \sum_{t=0}^{H} \gamma^t \left( \|\boldsymbol{z}_{t+1} - sg(E_{\phi}(\boldsymbol{s}_{t+1}, \boldsymbol{e}))\|_2^2 + CE(\hat{r}_t, r_t) \right) \right] \quad (10)$$

where $sg(\cdot)$ is the stop-gradient operator and CE is the cross-entropy loss function. Reward prediction is formulated as a discrete regression problem in log-transformed space. Furthermore, $E_{\phi}$ and $F_{\phi}$ use SimNorm activation (Eq. 4) in their output layers. All trainable models are fully-connected MLPs with LayerNorm Ba et al. (2016) and Mish activation Misra (2019). The complete algorithm is shown in Algorithm 1. Further implementation details can be found in Appendix C.

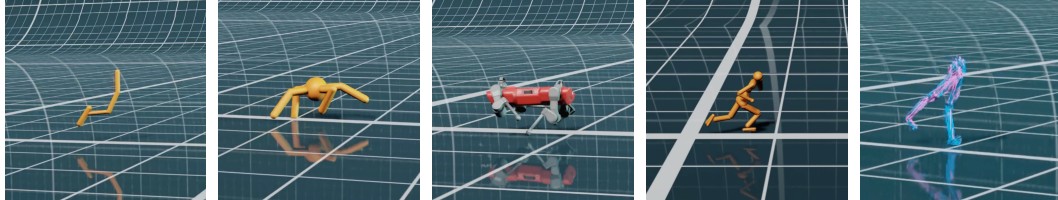

**Figure 4:** High-dimensional single-task environments (left to right): Hopper, Ant, Anymal, Humanoid and SNU Humanoid. Our method successfully learns tasks with up to $m = 152$ continuous action dimensions. Additional 80 multi-task environments used in this paper are listed in Appendix E

# 4 EXPERIMENTAL RESULTS

## 4.1 CONTACT-RICH SINGLE TASKS

The aim of this section is to understand whether smooth world models create better optimization landscapes than ground-truth dynamics, facilitating efficient FoG optimization. We study this on 5 continuous control tasks (Figure 4) with up to $\mathcal{A} = \mathbb{R}^{152}$ using the differentiable simulator dflex Xu et al. (2022). Comparisons include SHAC Xu et al. (2022), which uses ground-truth dynamics and rewards with a similar actor-critic architecture to PWM. Furthermore, we compare against two world model approaches. DreamerV3 Hafner et al. (2023) learns its world model via reconstruction, its actor via ZoG optimization, and critic via Model-based Value Expansion (MVE) Feinberg et al. (2018). TD-MPC2 Hansen et al. (2024) uses the same world model as PWM but learns a policy in a model-free fashion and actively plans at inference time. We additionally include model-free baselines PPO Schulman et al. (2017) and SAC Haarnoja et al. (2018). This comparison allows us to understand whether (1) FoG-based optimization can learn better policies asymptotically and (2) whether smooth world models induce better optimization landscapes for FoG optimization.

We conduct this experiment across 5 tasks with 10 seeds each, where PWM, DreamerV3, and TD-MPC2 use pre-trained world models and are left to learn a policy and fine-tune their world models online. This is done to enable a fair comparison to SHAC, which directly has access to the simulation model and does not require any training. The results in Figure 5 reveal that (1) PWM is able to learn policies with higher reward than SHAC asymptotically, indicating that regularized world models induce smoother optimization landscapes than the true (discontinuous) dynamics. Furthermore (2), despite using the same world model, our method is able to learn policies with higher rewards than TD-MPC2 without the need for online planning. However, PWM does not scale well to the highest-dimensional task. More experiment details and results are included in Appendix D.

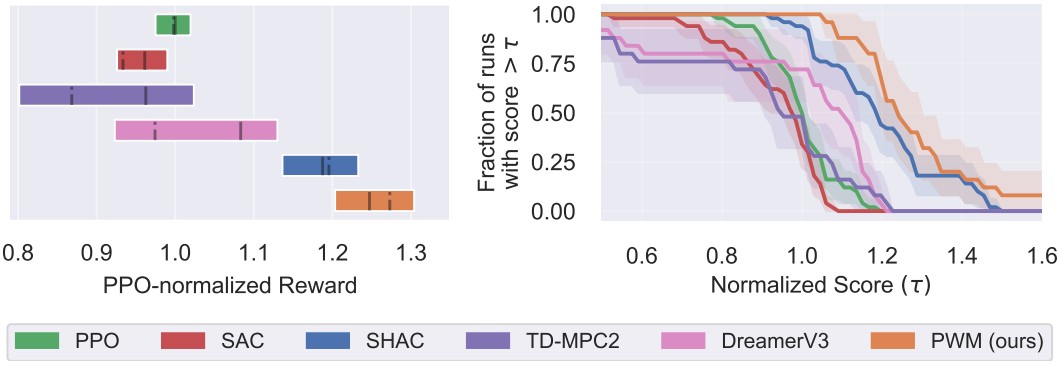

**Figure 5: Aggregate results from high-dimensional locomotion tasks** where each agent is trained to solve just that task (i.e. specialist). The **left** figure summarizes rewards achieved at the end of training using 50% IQM for the solid lines and 95% CI as suggested by Agarwal et al. (2021), as well as mean for the dashed lines. We see that PWM achieves higher rewards than our main baselines TD-MPC2 and SHAC. The **right** figure shows score distributions across all tasks which lets us understand the performance variability of each approach.

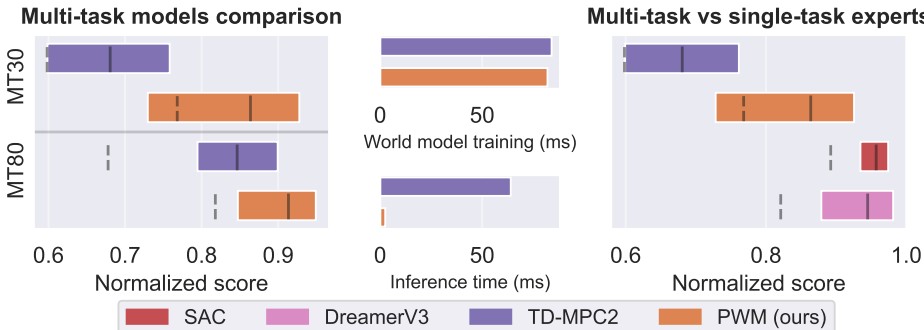

**Figure 6: Multi-task results.** The **left** figure shows results of multi-task agents in the 30 and 80 task set settings which include environments from dm_control Tunyasuvunakool et al. (2020) and MetaWorld Yu et al. (2020). The results show 50% IQM with the solid lines and mean with the dashed lines. The bars represent 95% CI. In both settings PWM achieves higher reward than TD-MPC2 without the need for online planning. The **middle** figure compares the training and inference times of TD-MPC2 and PWM for the 48M parameter model. PWM has significantly lower inference time as it does not plan online. The **right** figure shows a comparison between multi-task PWM and TD-MPC2 and single-task experts SAC and DreamerV3 on the MT30 task set. Notably, PWM is able to match the performance of SAC and DreamerV3.

## 4.2 MULTI-TASK WORLD-MODEL

We analyze the scalability of our proposed framework and method to large multi-task pre-trained world models. We evaluate on two settings: (1) 30 continuous control dm_control tasks Tunyasuvunakool et al. (2020) ranging from $m = 1$ to $m = 6$ and (2) 80 tasks, which include 50 additional manipulation tasks from MetaWorld Yu et al. (2020) with $n = 39$ and $m = 4$. These two multi-task settings were introduced as *MT30* and *MT80* by Hansen et al. (2024). In conducting our experiments, we harness the same data and world model architecture as TD-MPC2. The data consists of 120k and 40k trajectories per dm_control and MetaWorld task, respectively generated by 3 random seeds of TD-MPC2 runs. The world models we use are the 48M parameter models introduced in Hansen et al. (2024) with slight modifications to make them differentiable (Appendix C).

To train PWM, we first pre-train the world models on the dataset in a similar fashion to TD-MPC2, but with training $H = 16$ and $\gamma = 0.99$ for better first-order gradients, as highlighted in Section 3.2. Then we train a PWM policy on each particular task using the offline datasets for 10k gradient steps, which take 9.3 minutes on an Nvidia RTX6000 GPU. We evaluate task performance for 10 seeds for each task and aggregate results in Figure 6. We compare against TD-MPC2, which learns a multi-task policy while pre-training its world model and relies on online planning at inference. We can see that PWM learns behavior, achieving a higher reward than TD-MPC2 while also being significantly faster at inference time. While the fast per-task training is enabled by FoG optimization, we also find that training a single multi-task policy produces poor results, as shown in Appendix F. We further compare our multi-task PWM policy to online-trained single-task experts SAC Haarnoja et al. (2018) and DreamerV3 Hafner et al. (2023). Figure 6 reveals that multi-task PWM, while disadvantaged, performs comparably to the single-task experts without requiring any environment interaction and only training policies for $\leq 10$ minutes per task. Additional results in Appendix E.

## 4.3 ABLATIONS

We perform 4 ablations on the complex single task experiments in order to understand the nuances of first-order optimization through world models with PWM.

We increase the **contact stiffness** to be more realistic, but also more stiff contact gives gradients with high sample error Suh et al. (2022). We run the same experiment as Section 4.1, but only for the Hopper task, and present the aggregate results from 5 random seeds in Figure 7a, where we normalize rewards by the maximum reward achieved by PPO in Section 4.1. We see that while PPO and PWM rewards remain similar to prior results, SHAC performance decreases by 48%. This shows that regularized world models are robust to stiff contact models and thus more generalizable than differentiable simulations.

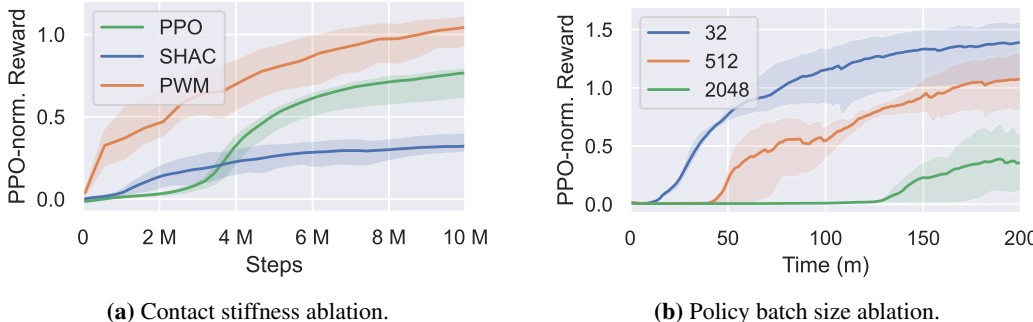

**(a)** Contact stiffness ablation.

**(b)** Policy batch size ablation.

**Figure 7: Left figure shows contact stiffness ablation** where we increase contact stiffness on the Hopper task and analyze the effects on policy learning. The results indicate that stiff (but realistic) contact has adverse effects on SHAC which uses the simulation model to learn. Meanwhile, PPO and PWM remain unaffected with PWM still obtaining 17% more reward than PPO asymptotically. The **right figure shows a policy batch size ablation** on the Any task where we vary only the batch size used to train the policy components of PWM. Unfortunately we observe that PWM provides best result within a unit of time by using small batch sizes. Both figures show 50% IQM and 95% CI over 5 random seeds.

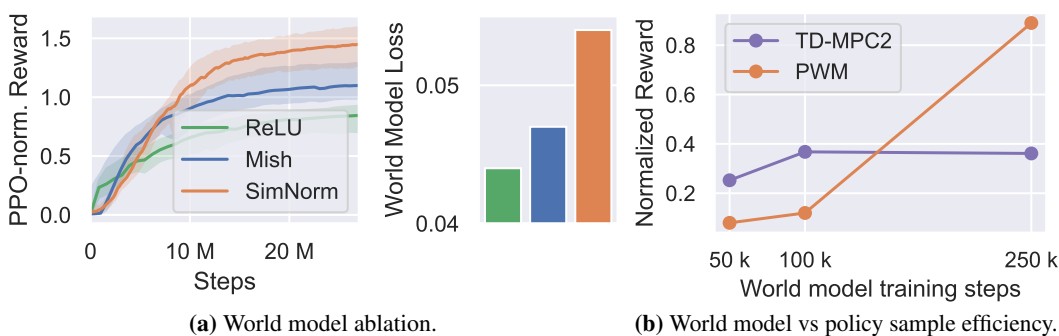

**(a)** World model ablation.

**(b)** World model vs policy sample efficiency.

**Figure 8:** The **left figure ablates the activation functions** of the world model used to learn policies on the Ant task. We progressively add more regularization to the world model via changes to the activation function and observe an inverse correlation between world model loss and policy reward. This indicates that we should not construct world models for accuracy but for policy learning. The **right figure investigates the policy sample efficiency** on 5 dm_control tasks. We use the same data to pre-train world models for varying amount of gradient steps and then train the policy for 50k gradient steps and compare against TD-MPC2 (without planning). The results indicate that PWM policies are significantly more sample efficient but also require better trained world models. All results shown are 50% IQM with 95% CI across 5 random seeds.

The **second ablation explores batch sizes** for policy learning with first-order gradients. Contrary to model-free methods, which can scale to large batch sizes, we find that FoG techniques like PWM benefit from smaller batch sizes. We explore this on the Ant task in Figure 7b, where we plot 50% IQM rewards over 5 random seeds. While larger batch sizes allow us to generate more data within a unit of time, that does not necessarily translate to learning better policies.

Next we **ablate the world model regularization**. We perform the same experiment as Section 4.1 on the Ant task but now pre-train 3 different world models. (1) with ReLU activation func., (2) with Mish activation func. and (3) with Mish activation func. and SimNorm activation func. at the output layers of $E_\phi$ and $F_\phi$. Figure 8a reveals that while less regularization results in lower world model error, that does not translate to learning better policies. Surprisingly, less regularized world models enable policies to start faster (up to 1M steps) but plateau to a suboptimal policy. Additional results in Appendix F.

To understand the **policy sample efficiency** of PWM while controlling for the world model, we perform an ablation where we pre-train the same world model for [50k, 100k, 250k] gradient steps on an offline dataset. Then we fix the world model and train only the policy components on the

same dataset for 50k gradient steps and measure the reward. We do this for 3 random seeds and 5 dm_control tasks. We repeat the same experiment for TD-MPC2 but disable its planning component in order to understand the learning dynamics of each method's policy components. The results in Figure 8b show that the PWM policy components are significantly more sample efficient than TD-MPC2 but also require better trained world models in order to obtain high reward.

## 5 RELATED WORK

**Reinforcement learning (RL)** strategies are divided into model-based and model-free approaches, with the latter not assuming a model of the environment Arulkumaran et al. (2017). Model-free approaches, such as PPO Schulman et al. (2017) and SAC Haarnoja et al. (2018) do not require a model of the environment and represent on-policy and off-policy methods, respectively. These algorithms use an actor-critic structure, where the critic assesses the policy while the actor updates it through gradient-based optimization to maximize rewards Konda & Tsitsiklis (1999).

**Gradient estimator types**. In the absence of direct access to dynamics and reward functions, it is common to use the Policy Gradient Theorem Sutton et al. (1999), a zeroth-order method, to estimate gradients. Although robust to discontinuities, this method exhibits high variance, leading to sample inefficiency Mohamed et al. (2020). In contrast, first-order gradients (FoG) offer lower variance by differentiating through the objective but struggle with discontinuities Suh et al. (2022). Differentiable simulations have risen as a tool to study the properties of gradient estimators Howell et al. (2022); Metz et al. (2021) and have produced model-based algorithms that use FoG optimization through physics to learn high-performing policies Xu et al. (2022); Georgiev et al. (2024).

**Multi-task models.** While traditional RL focuses on single-task policies, the broader robotics field is increasingly adopting large multi-task models through behavior cloning Firoozi et al. (2023). Recent efforts like Open X Padalkar et al. (2023) and Octo Octo Model Team et al. (2024) have demonstrated improved performance across various tasks and embodiments by leveraging large models and datasets. However, the potential of these large-scale approaches in RL remains largely unexplored. While GATO Reed et al. (2022) attempted to scale model-free RL across multiple tasks, it faced challenges with sample inefficiency and required significant fine-tuning. Conversely, TD-MPC2 Hansen et al. (2024) successfully scaled a 317M parameter world model for online planning across 80 tasks. While showing impressive multi-task scalability, it failed to solve all tasks and exhibits limited scalability due to online planning. Our work builds on the world model architecture proposed by TD-MPC2 but employs FoG optimization for policy learning and extracts per-tasks policies. DreamerV3 Hafner et al. (2023) also integrates world models with FoG but focuses on online learning without addressing multi-task scenarios. Our work delves deeper into the relationship between world models and policy learning, exploring the essential characteristics of world models that facilitate efficient optimization.

## 6 CONCLUSION

In this study, we analyzed world models through policy gradient estimation and identified an inverse correlation between the accuracy of world models and episode rewards. We concluded that world models should prioritize smoothness and a smaller optimality gap over accuracy to enhance policy performance. Building on these insights, we propose Policy learning through Multi-task World Models (PWM), a MBRL algorithm that integrates smooth world models with first-order gradient (FoG) optimization. Our evaluations showed that PWM can outperform existing methods, including those with access to ground-truth simulation dynamics, in learning high-reward policies for high-dimensional tasks. To scale to a multi-task settings, we propose a framework where world models are pre-trained offline and treated as differentiable simulations. Our results demonstrate that PWM can be used to learn expert policies in <10 minutes per task, achieving higher rewards without the need for expensive online planning. With ample data and large, smooth world models, we believe this approach has significant potential for scalability.

**Limitations.** Despite its demonstrated efficacy, PWM has notable limitations. Firstly, performance relies heavily on the availability of substantial pre-existing data to train the world model, which might not always be feasible, especially in novel or low-data environments. Secondly, although PWM facilitates fast and cost-effective policy training, it necessitates re-training for each new task, which could limit its applicability in scenarios requiring rapid adaptation to diverse tasks. Lastly, the current TD-MPC2 world models used are difficult to train at scale due to their auto-regressive formulation.

**Reproducibility statement.** Code, training data and checkpoints are made available at imgeorgiev.com/pwm. We rely on dflex, MetaWorld, DMControl and MuJoCo for simulation which are publicly available under MIT and Apache 2.0 licenses. We use multi-task data from TD-MPC2 which is publicly available. Implementation details and full list of hyper-parameters are available in Appendix C.

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

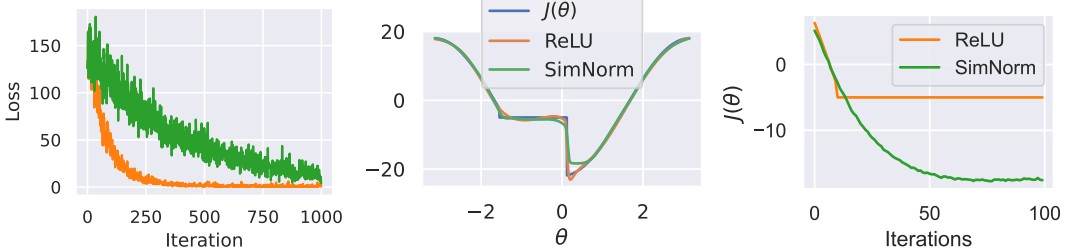

**Figure 10: Extended ball-wall toy example results.** The **left** figure shows the model losses as they are trained to approximate the target function $J(\theta)$. The middle figure shows the output of the trained model across the spectrum of $\theta$ as well as the true target. This is the function we attempt to minimize in the **right** figure. We can see that when using the MLP with ReLU activation functions, the optimizer quickly gets stuck in local minima while the model using SimNorm activation function is able to find a solution closer to the true one.

## A   BALL-WALL EXAMPLE DETAILS

This section provides more details on the ball-wall example used to showcase the issues of optimizing through contact in Section 3.1. In constructing this toy example we chose a simple physical system that exhibits contact discontinuities. Inspired by Suh et al. (Suh et al., 2022), we constructed a simple problem of a point mass (ball) being thrown forward (x direction) at a fixed velocity $v$. The optimization parameter of interest is the initial angle $\theta$ and the goal is to maximize forward distance traveled (in 2D). For simplicity we assume that the ball sticks to the wall (without complex contact) which can be expressed as:

$$J(\theta) = x_t = f(\theta) = \begin{cases} x_0 + v\cos(\theta)t + \frac{1}{2}gt^2 & \text{if } y_{\text{contact}} > h \\ w & else \end{cases} \quad (11)$$

where $g = 9.81$ is gravity, $h$ and $w$ are the height and width of the wall, $(x_0, y_0)$ is the starting position, $v = 10$ is the starting velocity and $t = 2$ is time. $y_{\text{contact}}$ is the height at the time of contact $t_{\text{contact}}$ which are both given by solving Eq. 11 for $f(\theta) = w$:

$$t_{\text{contact}} = \frac{-v\cos(\theta) + \sqrt{v^2\cos^2(\theta) + aw}}{a} \qquad y_{\text{contact}} = y_0 + v\sin(\theta)t + \frac{1}{2}gt^2$$

We visualize the toy example in Figure 9 to aid reading. With $f(\theta)$ defined, we attempt to learn it with two Multi-Layer Perceptrons (MLPs). We configure them to have 2 hidden layers of 32 neurons each. The first MLP uses ReLU activation functions, while the latter uses SimNorm activation functions as defined in Eq. 4. Both models are initialized with identical random parameters and are trained with the ADAM optimizer with learning rate $\alpha = 2 \times 10^{-3}$ for 100 epochs using a batch size of $B = 50$. The data we use to train the models was 1000 uniform samples of $f(\theta)$ within $\theta \in [-\pi, \pi]$. Figure 10 shows the training losses of the models, induced optimization landscapes and the losses when attempting to maximize the models as you would do in an RL setting.

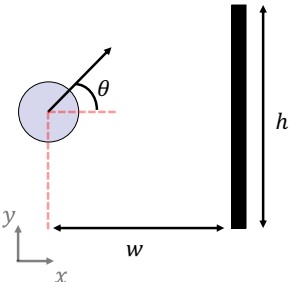

**Figure 9:** Pedagogical ball-wall toy problem visualized.

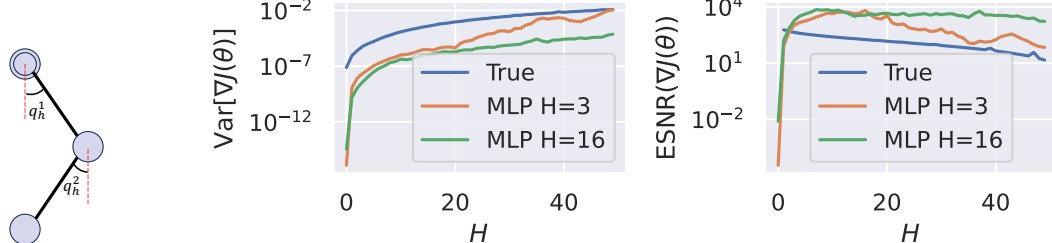

**Figure 11: Double pendulum pedagogical example.** The middle figure evaluates the variance of policy gradient estimates over $N = 100$ Monte-Carlo samples for varying horizons $H$. The right figure shows the same data but plots the Expected Signal-to-Noise ratio (ESNR) with higher values translating to more useful gradients. These results suggests that world models trained over long horizon trajectories provide more useful gradients.

## B  DOUBLE PENDULUM EXAMPLE DETAILS

The double pendulum (also known as Acrobot (Murray & Hauser, 1991)) is a classic under-actuated chaotic system. It is characterized by its sensitivity to initial conditions where even small perturbations result in large gradient variance with long horizon ($> 20$) trajectories. We chose this system to analyze variance and expected signal-to-noise ratio (ENSR) in Section 3.2 as it is the easiest problem exhibiting chaosness. We model this toy problem similar to DMControl (Tunyasuvunakool et al., 2020) in our differentiable simulator, dflex as we need ground truth gradients for comparison. The first link with angle $\theta_1$ is fixed to the base and not actuated. The second link with angle $\theta_2$ is the only control input via $\dot{\theta}_2$. The state of the system id calculated as:

$$\boldsymbol{s} = [\cos(\theta_1), \sin(\theta_1), \cos(\theta_2), \sin(\theta_2), \dot{\theta}_1, \dot{\theta}_2]$$

The objective of this toy example is to bring and balance the pendulum upwards which we achieve by formulating a reward:

$$r(\boldsymbol{s}, \boldsymbol{a}) = -\theta_1^2 - \theta_2^2 - 0.1\dot{\theta}_2^2$$

Next we train world models to approximate the dynamics and reward above. For this we collect data with the SHAC algorithm (Xu et al., 2022) over 3 different runs for a total of 24,000 episodes of length 240 timesteps. Maximum episode reward achieved during data collection was -942.95. Then we train two TDMPC2 (Hansen et al., 2024) world models on the collected data. We use the 5M parameter model which features a latent state of dimension of 512, encoder $E_\phi$ with one hidden dimension of 256, dynamics model MLP with 2 hidden layers with 512 neurons and a rewards model of the same design. We keep the same hyper-parameters as per the origin work by Hensen et al. but use $\gamma = 0.99$ which we found to reduce variance substantially. We train two models with different training horizons $H = 3$ and $H = 16$ for 100k batch samples and a batch size of 1024.

With the trained models, we now compare the variance of stochastic gradients provided by the true dynamics of the simulation and the two trained models. We do this by loading the best policy learned by SHAC during data collection and executing a $H = 50$ rollout across the 3 models. We ensure that the same actions are taken for each evaluated models and collect 100 Monte Carlo samples. In addition to variance, we report ESNR as suggested by (Parmas et al., 2023) and defined in 5. Higher ESNR translate to more useful gradients and we naturally should expect values to decrease with increased $H$. We reported the results in Figure 3 but also duplicate them in Figure 11 for convenience and ease of reading.

## C IMPLEMENTATION DETAILS AND HYPER-PARAMETERS

The section details several implementation details of PWM that we thought are not crucial for understanding the proposed approach in Section 3.3 but are important for replicating the results.

1. **Reward binning** - the reward model we use in PWM is formulated as a discrete regression problem where $\mathbb{R}$ rewards are discretized into a predefined number of bins. Similar to (Hansen et al., 2024; Lee et al., 2023), we do this to enable robustness to reward scale and multi-task-ness. In particular, we perform two-hot encoding using SymLog and SymExp operators which are mathematically defined as:

$$\mathrm{SymLog}(x) = \mathrm{sign}(x)\log(1 + |x|) \qquad \mathrm{SymExp}(x) = \mathrm{sign}(x)(e^{|x|} - 1)$$

Two-hot encoding is then performed with:

```
def two_hot(x):
    x = clamp(symlog(x), vmin, vmax)
    bin_idx = floor((x - vmin) / bin_size)
    bin_offset = (x - vmin) / bin_size - bin_idx
    soft_two_hot = zeros(x.size(0), num_bins)
    soft_two_hot[bin_idx] = 1 - bin_offset
    soft_two_hot[bin_odx + 1] = bin_offset
    return soft_two_hot
```

Inverting this operation to get back to scalar rewards would usually involve $\mathrm{SymExp}(x)$ but note that the $\mathrm{sign}(x)$ operator is not differentiable and would therefore not work for FoG. Instead, we chose to omit the $\mathrm{SymExp}(x)$ operation which technically now returns pseudo-rewards but also gradients which we found sufficient for policy learning:

```
def two_hot_inversion(x):
    vals = linspace(vmin, vmax, num_bins)
    x = softmax(x)
    x = torch.sum(x * vals, dim=-1)
    return x
```

2. **Critic training** - while Algorithm 1 function to similar results as presented in 4, we found it beneficial to split the critic training data from a single rollout into several smaller mini-batches and over them for multiple gradient steps. In our implementation we split the data into 4 mini-batches and perform 8 gradient steps over them with uniform sampling. With a $H = 16$ and batch size 64, this translates to a critic batch size of 256.

3. **Minimum policy noise** - Due to the larger amount of gradient steps needed, we noticed that PWM's actor tends to collapse to a deterministic policy rapidly. As such, we found it beneficial to include a lower bound on the standard deviation of the action distribution in order to maintain stochasticity in the optimization process. We have used $0.24$ throughout this paper. While similar results would be possible by adding an entropy term (Schulman et al., 2017), we found our current solution sufficient

4. **World model fine-tuning** - Throughout all of our experiments we found that the offline data used to train PWM's world model to be crucial to learning a good policy. In very high-dimensional tasks such as Humanoid SNU, collecting extensive data is a difficult task. As such, in these tasks we found it beneficial to online fine-tune the world model. We do this on all single-task experiments of Section 4.1 using the default hyper-parameters and a replay buffer of size 1024.

| Hyper-parameter | Value |
| --- | --- |
| **Policy components** | |
| Horizon ($H$) | 16 |
| Batch size | 64 |
| $\alpha_{\boldsymbol{\theta}}$ | $5 \times 10^{-4}$ |
| $\alpha_{\psi}$ | $5 \times 10^{-4}$ |
| Actor grad norm | 1 |
| Critic grad norm | 100 |
| Actor hidden layers | $[400, 200, 100]$ |
| Critic hidden layers | $[400, 200]$ |
| Number of critics | 3 |
| $\lambda$ | 0.95 |
| $\gamma$ | 0.99 |
| Critic batch split | 4 |
| Critic iterations | 8 |
| **World model components (48M)** | |
| Latent state ($\boldsymbol{z}$) dimension | 768 |
| Horizon ($H$) | 16 |
| Batch size | 1024 |
| $\alpha_{\phi}$ | $3 \times 10^{-4}$ |
| World model grad norm | 20.0 |
| SimNorm $V$ | 8 |
| Reward bins | 101 |
| Encoder $E_{\phi}$ hidden layers | $[1792, 1792, 1792]$ |
| Dynamics $F_{\phi}$ hidden layers | $[1792, 1792]$ |
| Reward $R_{\phi}$ hidden layers | $[1792, 1792]$ |
| Task encoding dimension | 96 |

**Table 1:** Table of hyper-parameters used in PWM, shared across all tasks.

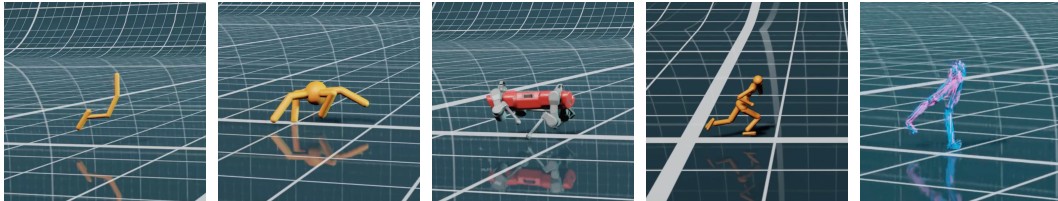

**Figure 12:** Locomotion environments (left to right): Hopper, Ant, Anymal, Humanoid and SNU Humanoid.

## D  CONTACT-RICH SINGLE TASK EXPERIMENT DETAILS

In Section 4.1, we explore 5 locomotion tasks with increasing complexity. They are described below and shown in Figure 4.

1. **Hopper**, a single-legged robot jumping only in one axis with $n = 11$ and $m = 3$.

2. **Ant**, a four-legged robot with $n = 37$ and $m = 8$.

3. **Anymal**, a more sophisticated quadruped with $n = 49$ and $m = 12$ modeled after (Hutter et al., 2016).

4. **Humanoid**, a classic contact-rich environment with $n = 76$ and $m = 21$ which requires extensive exploration to find a good policy.

5. **SNU Humanoid**, a version of Humanoid lower body where instead of joint torque control, the robot is controlled via $m = 152$ muscles intended to challenge the scaling capabilities of algorithms.

All tasks share the same common main objective - maximize forward velocity $v_x$:

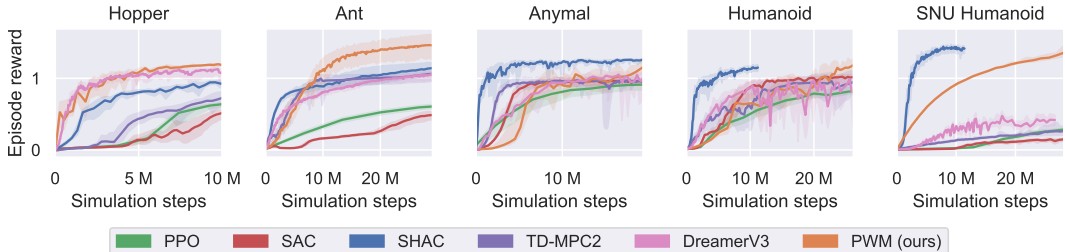

**Figure 13: Learning curves for each environment**. This figure shows 50% IQM and 95% CI across 10 random seeds for each task in the dflex simulation suite. Rewards are normalized by the maximum reward achieved by PPO (usually $\geq$ 100M steps). While PWM remains competitive with SHAC for most tasks, it does not scale well to the 152 action dimension SNU Humanoid. Note that SHAC uses gradients from the differentiable simulation directly and converges in fewer samples, explaining the truncated curves.

| Environment | Reward |
|---|---|
| Hopper | $v_x + R_{height} + R_{angle} - 0.1\|\boldsymbol{a}\|_2^2$ |
| Ant | $v_x + R_{height} + 0.1R_{angle} + R_{heading} - 0.01\|\boldsymbol{a}\|_2^2$ |
| Anymal | $v_x + R_{height} + 0.1R_{angle} + R_{heading} - 0.01\|\boldsymbol{a}\|_2^2$ |
| Humanoid | $v_x + R_{height} + 0.1R_{angle} + R_{heading} - 0.002\|\boldsymbol{a}\|_2^2$ |
| Humanoid STU | $v_x + R_{height} + 0.1R_{angle} + R_{heading} - 0.002\|\boldsymbol{a}\|_2^2$ |

**Table 2:** Rewards used for each task bench-marked in Section 4

We additionally use auxiliary rewards $R_{height}$ to incentivise the agent to, $R_{angle}$ to keep the agents' normal vector point up, $R_{heading}$ to keep the agent's heading pointing towards the direction of running and a norm over the actions to incentivise energy-efficient policies. For most algorithms, none of these rewards apart from the last one are crucial to succeed in the task. However, all of them aid learning policies faster.

$$R_{height} = \begin{cases} h - h_{term} & if\, h \geq h_{term} \\ -200(h - h_{term})^2 & if\, h < h_{term} \end{cases}$$

$$R_{angle} = 1 - \left(\frac{\theta}{\theta_{term}}\right)^2$$

$R_{angle} = \|\boldsymbol{q}_{forward} - \boldsymbol{q}_{agent}\|_2^2$ is the difference between the heading of the agent $\boldsymbol{q}_{agent}$ and the forward vector $\boldsymbol{q}_{agent}$. $h$ is the height of the CoM of the agent and $\theta$ is the angle of its normal vector. $h_{term}$ and $\theta_{term}$ are parameters that we set for each environment depending on the robot morphology. Similar to other high-performance RL applications in simulation, we find it crucial to terminate episode early if the agent exceeds these termination parameters. However, it is worth noting that AHAC is still capable of solving all tasks described in the paper without these termination conditions, albeit slower.

All results presented in Figure 13 are for 10 random seeds using the simulator in a vectorized fashion with 64 parallel environments for all approaches, except PPO which uses 1024. We note that while simulation steps appear high, all of these experiments are executed $\leq$ 2 hours on an Nvidia RTX6000 GPU. In addition the the learning curves of Figure 13, we also present tabular results below:

We note that TDMPC2 and PWM use pre-trained world models on 20480 episodes of each task. The world models are trained for 100k gradient steps and the same world models (specific to each task) are loaded into both approaches. The data consists of trajectories of varying policy quality generated with the SHAC algorithm. Trajectories include near-0 episode rewards as well as the highest reward achieved by SHAC. Note that we also run an early termination mechanism in these tasks which is done to accelerate learning and iteration.

|  | Hopper | Ant | Anymal | Humanoid | SNU Humanoid |
|---|---|---|---|---|---|
| PPO | $1.00 \pm 0.11$ | $1.00 \pm 0.12$ | $1.00 \pm 0.03$ | $1.00 \pm 0.05$ | $1.00 \pm 0.09$ |
| SAC | $0.87 \pm 0.16$ | $0.95 \pm 0.08$ | $0.98 \pm 0.06$ | $1.04 \pm 0.04$ | $0.88 \pm 0.11$ |
| DreamerV3 | $1.15 \pm 0.47$ | $1.12 \pm 0.45$ | $1.18 \pm 0.47$ | $1.03 \pm 0.43$ | $0.48 \pm 0.21$ |
| TDMPC2 | $0.85 \pm 0.37$ | $1.07 \pm 0.44$ | $0.98 \pm 0.48$ | $1.05 \pm 0.46$ | $0.26 \pm 0.12$ |
| SHAC | $1.02 \pm 0.03$ | $1.16 \pm 0.13$ | $1.26 \pm 0.04$ | $1.15 \pm 0.04$ | $1.44 \pm 0.08$ |
| PWM | $1.20 \pm 0.29$ | $1.46 \pm 0.31$ | $1.16 \pm 0.24$ | $1.19 \pm 0.025$ | $1.36 \pm 0.56$ |

**Table 3:** Tabular results of the asymptotic (end of training) rewards achieved by each algorithm across all tasks. The results presented are PPO-normalised 50 % IQM and standard deviation across 10 random seeds. Most algorithms have been trained until convergence.

|  | Hopper | Ant | Anymal | Humanoid | SNU Humanoid |
|---|---|---|---|---|---|
| PPO | $4742 \pm 521$ | $6605 \pm 793$ | $12029 \pm 360$ | $7293 \pm 365$ | $4114 \pm 370$ |
| SAC | $4126 \pm 759$ | $6275 \pm 528$ | $11788 \pm 722$ | $7285 \pm 292$ | $3620 \pm 453$ |
| DreamerV3 | $5436 \pm 2213$ | $7370 \pm 3002$ | $14169 \pm 5692$ | $7546 \pm 3106$ | $2018 \pm 873$ |
| TDMPC2 | $4027 \pm 1768$ | $7080 \pm 2885$ | $11787 \pm 4702$ | $7634 \pm 3317$ | $1121 \pm 525$ |
| SHAC | $4837 \pm 142$ | $7662 \pm 859$ | $15157 \pm 481$ | $8387 \pm 292$ | $5924 \pm 329$ |
| PWM | $5680 \pm 2303$ | $9672 \pm 2012$ | $13927 \pm 2882$ | $8661 \pm 1792$ | $5767 \pm 2394$ |

**Table 4:** Tabular results of the asymptotic (end of training) rewards achieved by each algorithm across all tasks. The results presented are 50 % IQM and standard deviation across 10 random seeds. All algorithms have been trained until convergence.

# E  MULTI-TASK EXPERIMENTS ADDITIONAL RESULTS

In this section we provide additional results on multi-task experiments. While we find it beneficial to train the world model at the same horizon as the policy learning, it is not strictly necessary to achieve good performance. In Figure 14 we present an ablation where we compare PWM world models pre-trained on horizons $H = 3$ and $H = 16$ and policies trained only with $H = 16$. These results reveal that $H = 16$ trained world models have only marginally higher scores. On deeper inspection, most of increased scores come form dm_control tasks which are harder than MetaWorld tasks on average. Therefore if training new world models, we advise using higher $H$; however if other pre-trained world models exist with suboptimal $H$, they will probably be also useful.

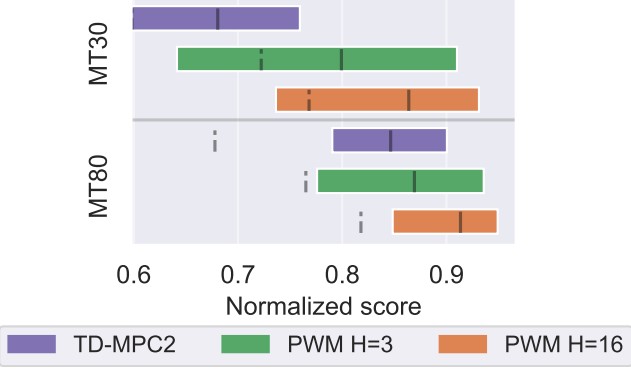

**Figure 14:** Horizon ablation of the world model

Figures 15 and 16 give scores for individual tasks for TDMPC2 and PWM across both the MT30 and MT80 task sets. We can observe that most of the increased performance of PWM is in dm_control tasks which are on average more difficult than MetaWorld.

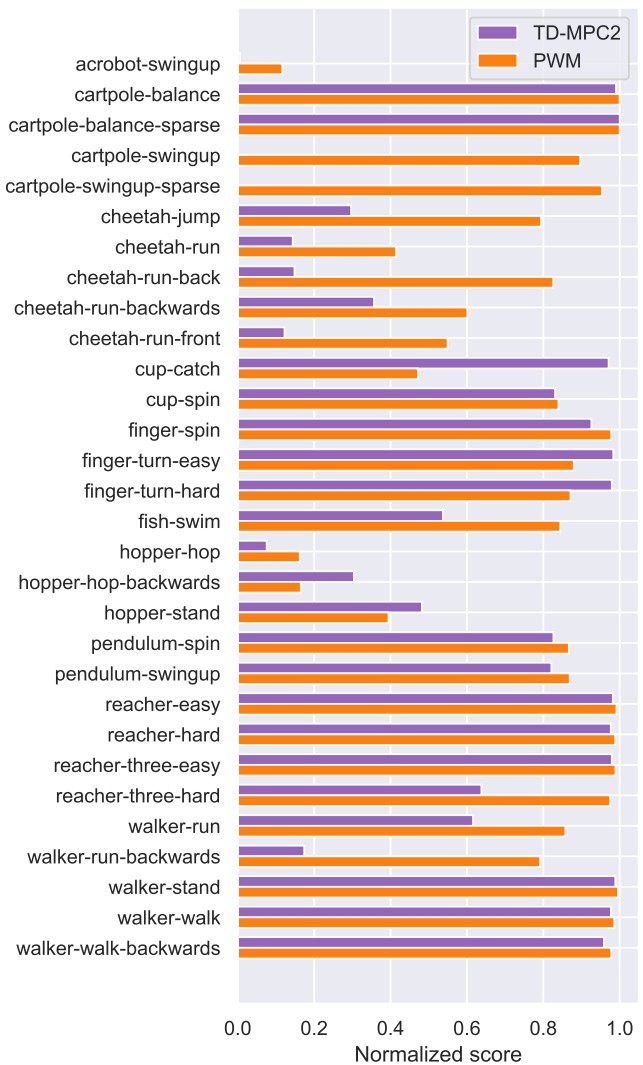

**Figure 15:** Individual task results for MT30 task set.

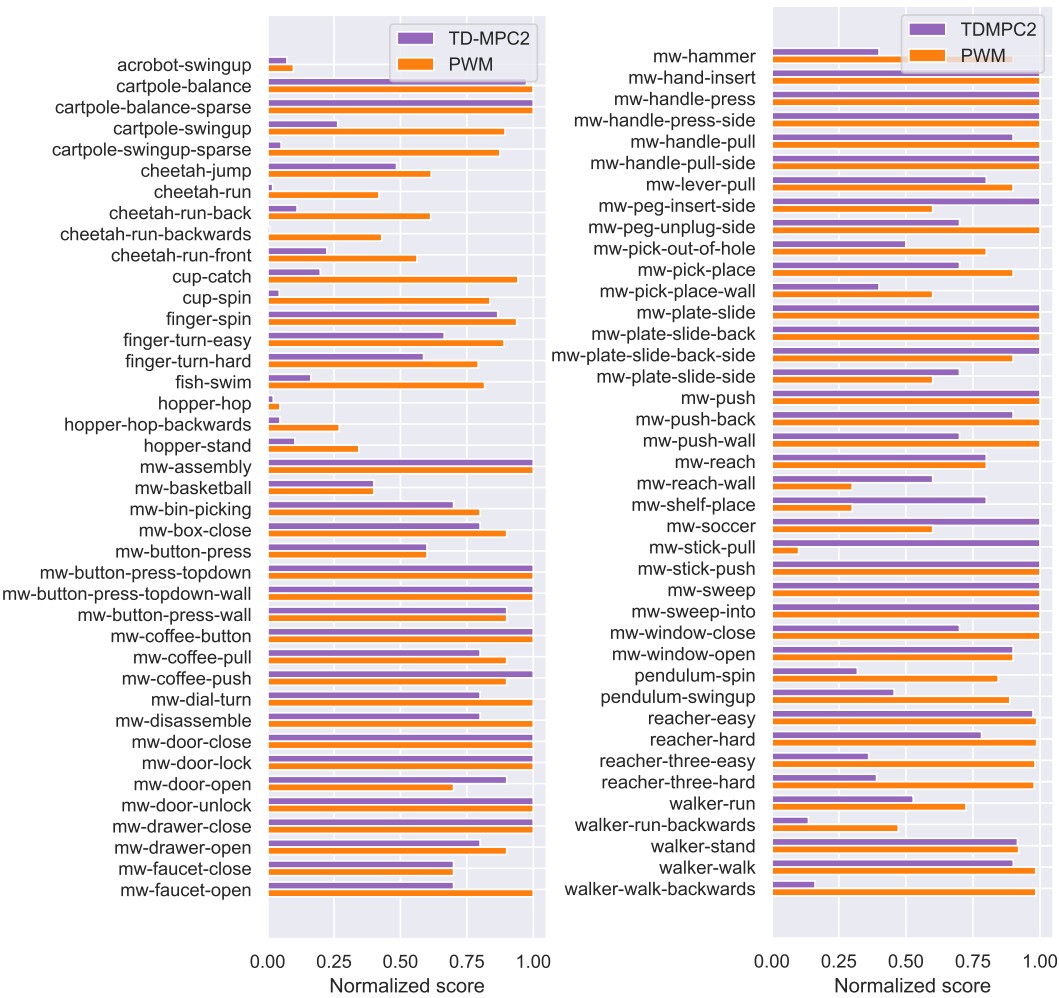

**Figure 16:** Individual task results for MT80 task set.

## F  ADDITIONAL ABLATION RESULTS

### F.1  WORLD MODEL REGULARIZATION ABLATION

We extend our world model regularization ablation from Section 4.3 to 3 additional dflex locomotion tasks - Hopper, Anymal and Humanoid. The complete results are presented in Figure 17 and follow the same format as the original results in Figure 8a - 50% IQM with 95% CI across 3 seeds. From the results we can observe that progressively adding more regularization, results in higher policy rewards. This follows the ablation results in our paper and the one of the core contribution of work - more accurate world models, do not results in better rewards. Instead of building world models for accuracy, we should build them to enable better policy optimization and thus higher rewards.

### F.2  TRAINING A MULTI-TASK POLICY

One of the contributions of our work is the multi-task learning framework of PWM. Instead of training a single multi-task policy, we propose training a single multi-task world model with a supervised objective and then extracting RL policies from it very efficiently. In this section we justify this empirically by reproducing the MT30 experiment of Section 4.2 with two additional baselines - TD-MPC2 without planning and PWM with a single multi-task policy. This strips down both algorithms to be very similar, the major difference being that TD-MPC2 features an off-policy SAC-like policy while PWM features an on-policy SHAC-like policy. Both are fed the same one-hot

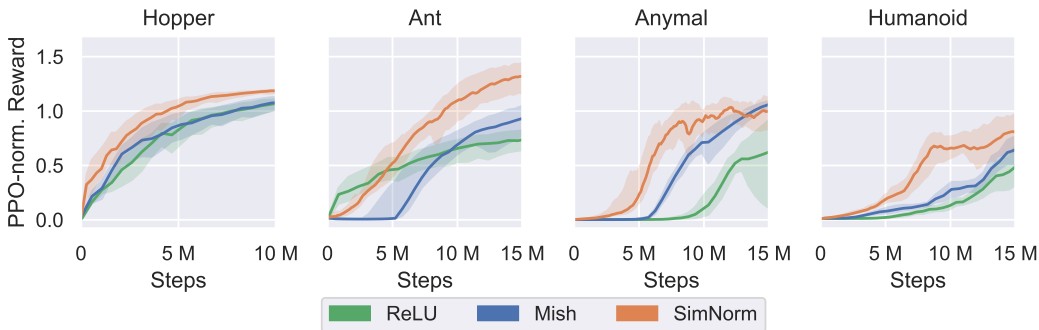

**Figure 17: World model regularization ablation.** This figure extends the ablation results of Figure 8a with additional tasks. The figure shows 50 % IQM with 95% CI over 5 seeds per task.

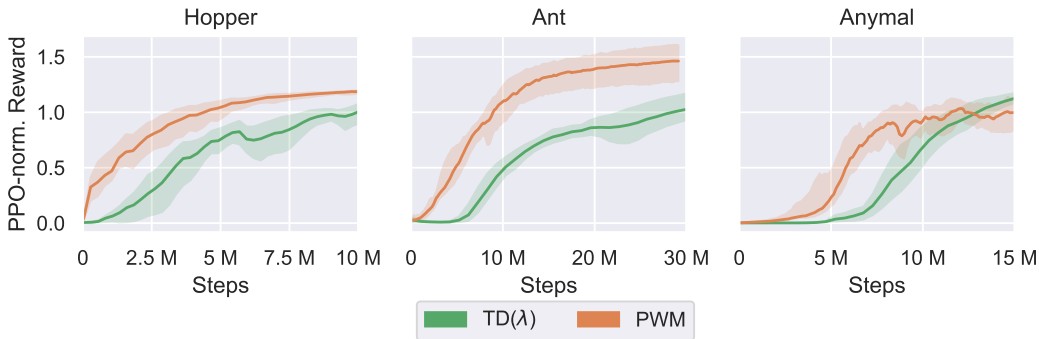

**Figure 19:** TD($\lambda$) ablation. This figure compares vanilla PWM with a version that uses a TD($\lambda$) actor objective. Results shown are 50% IQM with 95% CI over 3 seeds.

task embeddings. The results in 18 show that both fail at solving the MT30 benchmark. In the case of TD-MPC2, this reveals its reliance on planning and in the case of PWM, this reveals the need to train policies per-task. We believe this is due to the unstable optimization objective of the RL problem.

### F.3 TD($\lambda$) ABLATION

Here we replace the actor objective of PWM with TD($\lambda$) similar to Dreamer (Hafner et al., 2019). We evaluate this on 3 single-task experiments - Hopper, Ant and Anymal and show the results in 19. We can see that PWM overall achieves higher rewards, which we believe is to more simple and less noisy gradients obtained via TD(N). Additionally, we found that TD($\lambda$) uses approx. 10% more computation due to having to compute more gradients. However, it is worth noting that TD($\lambda$) learning curves appear more stable with higher dimensional tasks such as Anymal and we believe this ablation requires more large-scale studying.

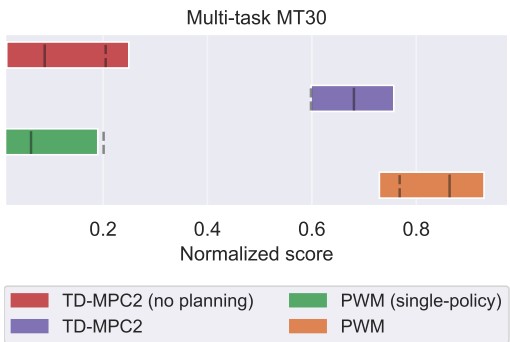

**Figure 18:** Framework ablation ablation. Extends the results of Figure 6 with two additional baselines: TD-MPC2 without planning and PWM with a single multi-task policy learned over all embeddings. The figure shows 50% IQM results with solid lines, mean with dashed lines and 95 % CI.

