# PWM: POLICY LEARNING WITH MULTI-TASK WORLD MODELS

## ABSTRACT

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

This underscores the efficacy of PWM and supports our broader contributions:

1. Through pedagogical examples and ablations, we show that more accurate world models do not result in better policies. Instead of pursuing world model improvements in isolation, we should aim to build world models that result in better policies.

2. When regularized correctly, world models enable efficient first-order optimization. We show that this results in better performing policies and faster training times in comparison to zeroth-order methods.

3. We propose PWM, a model-based algorithm for learning continuous control policies from pre-trained multi-task world models that can solve tasks in <10 minutes using FoG optimization.

## 2 BACKGROUND

We focus on discrete-time and infinite horizon Reinforcement Learning (RL) scenarios characterized by system states $\boldsymbol{s} \in \mathbb{R}^n = \mathcal{S}$, actions $\boldsymbol{a} \in \mathbb{R}^m = \mathcal{A}$, dynamics function $f : \mathcal{S} \times \mathcal{A} \to \mathcal{S}$ and a reward function $r : \mathcal{S} \times \mathcal{A} \to \mathbb{R}$. Combined, these form a Markov Decision Problem (MDP) summarized by the tuple $(\mathcal{S}, \mathcal{A}, f, r, \gamma)$ where $\gamma$ is the discount factor. Actions at each timestep $t$ are sampled from a stochastic policy $\boldsymbol{a}_t \sim \pi_\theta(\cdot|\boldsymbol{s}_t)$, parameterized by $\boldsymbol{\theta}$. The goal of the policy is to maximize the cumulative discounted rewards:

$$\max_{\boldsymbol{\theta}} J(\boldsymbol{\theta}) := \max_{\boldsymbol{\theta}} \mathbb{E}_{\substack{\boldsymbol{s}_1 \sim \rho(\cdot) \\ \boldsymbol{a}_t \sim \pi_\theta(\cdot|\boldsymbol{s}_t)}} \left[ \sum_{t=1}^{\infty} \gamma^t r(\boldsymbol{s}_t, \boldsymbol{a}_t) \right] \tag{1}$$

where $\rho(\boldsymbol{s}_1)$ is the initial state distribution. Since this maximization over an infinite sum is intractable, in practice we often maximize over a value estimate. The value of a state $\boldsymbol{s}_t$ is defined as the expected reward follow the policy $\pi_\theta$

$$V_{\boldsymbol{\psi}}^{\pi}(\boldsymbol{s}_t) := \mathbb{E}_{\boldsymbol{a}_h \sim \pi_\theta(\cdot|\boldsymbol{s}_h)} \left[ \sum_{h=t}^{\infty} \gamma^h r(\boldsymbol{s}_h, \boldsymbol{a}_h) \right] \tag{2}$$

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

$$\max_{\boldsymbol{\theta}} J(\boldsymbol{\theta}) := \max_{\boldsymbol{\theta}} \mathbb{E}_{\substack{\boldsymbol{s}_1 \sim \rho(\cdot) \\ \boldsymbol{a}_t \sim \pi_\theta(\cdot|\boldsymbol{s}_t)}} \left[ \sum_{t=1}^{\infty} \gamma^t r(\boldsymbol{s}_t, \boldsymbol{a}_t) \right] \tag{1}$$

where $\rho(\boldsymbol{s}_1)$ is the initial state distribution. Since this maximization over an infinite sum is intractable, in practice we often maximize over a value estimate. The value of a state $\boldsymbol{s}_t$ is defined as the expected reward follow the policy $\pi_\theta$

$$V_{\boldsymbol{\psi}}^{\pi}(\boldsymbol{s}_t) := \mathbb{E}_{\boldsymbol{a}_h \sim \pi_\theta(\cdot|\boldsymbol{

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

We train the MLPs and observe the smoothing effects of the learned models in Figure 2b. While the MLP smooths the problem landscape, it also introduces a local minimum when attempting to optimize with gradient descent starting from (e.g.) $\theta = -\pi$, leading to a large optimality gap (difference between the solution and the optimal solution: $\|\hat{\theta} - \theta^*\|$). In contrast, the SimNorm MLP has additional regularization which reduces the optimality hap, at the expense of model accuracy (Table 2c). This inverse correlation between the optimality gap and model error is known as objective mismatch (Lambert et al., 2020). Therefore, we believe that regularized learned models can reduce gradient sample error, and thus the optimality gap, enabling more efficient FoG optimization in non-smooth environments. Further details

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

 $\psi \leftarrow \psi - \alpha_{\psi} \mathcal{L}_V(\psi)$  ▷ Eq. 7-9
**end**

---

$$\mathcal{L}_{\pi}(\boldsymbol{\theta}) := \mathbb{E}_{\substack{\boldsymbol{s}_1 \sim \rho(\cdot) \\ \boldsymbol{a}_h \sim \pi_{\boldsymbol{\theta}}(\cdot|\boldsymbol{z}_h)}} \left[ \sum_{h=1}^{H-1} \gamma^h R_{\phi}(\boldsymbol{z}_h, \boldsymbol{a}_h) + \gamma^H V_{\psi}(\boldsymbol{z}_H) \right] \quad \text{where} \quad \begin{matrix} \boldsymbol{z}_1 = E_{\phi}(\boldsymbol{s}_1) \\ \boldsymbol{z}_{t+1} = F_{\phi}(\boldsymbol{z}_t, \boldsymbol{a}_t) \end{matrix} \quad (6)$$

The critic is trained in a model-free fashion using TD($\lambda$) over an $H$-step rollout in latent space $\boldsymbol{z}$ as seen in other similar on-policy methods (Sutton & Barto, 2018; Hafner et al., 2019; Xu et al., 2022):

$$V_h(\boldsymbol{z}_t) := \sum_{n=t}^{t+h-1} \gamma^{n-t} R_{\phi}(\boldsymbol{z}_n, \boldsymbol{a}_n) + \gamma^{t+h} V_{\psi}(\boldsymbol{z}_{t+h}) \quad (7)$$

$$\hat{V}(\boldsymbol{z}_t) := (1 - \lambda) \left[ \sum_{h=1}^{H-t-1} \lambda^{h-1} V_h(\boldsymbol{z}_t) \right] + \lambda^{H-t-1} V_H(\boldsymbol{z}_t) \quad (8)$$

$$\mathcal{L}_V(\psi) := \sum_{h=t}^{t+H} \left\| V_{\psi}(\boldsymbol{z}_h) - \hat{V}(\boldsymbol{z}_h) \right\|_2^2 \quad (9)$$

We use an ensemble of 3 critics to reduce variance. To enable FoG optimization, it is important to use a well-regularized world model. We use the $\left( E_{\phi}(\boldsymbol{s}, \boldsymbol{e}), F_{\phi}(\boldsymbol{s}, \boldsymbol{a}, \boldsymbol{e}), R_{\phi}(\boldsymbol{s}, \boldsymbol{a}, \boldsymbol{e}) \right)$ model proposed by TD-MPC2 (Hansen et al., 2024) with learnable task embeddings $\boldsymbol{e}$. It is trained in an auto-regressive fashion by sampling data from a buffer with loss function:

$$\mathcal{L}_{wm}(\phi) = \mathbb{E}_{(\boldsymbol{s}, \boldsymbol{a}, r, \boldsymbol{s}', \boldsymbol{e})_{0:H} \sim \mathcal{B}} \left[ \sum_{t=0}^{H} \gamma^t \left( \|\boldsymbol{z}_{t+1} - sg(E_{\phi}(\boldsymbol{s}_{t+1}, \boldsymbol{e}))\|_2^2 + CE(\hat{r}_t, r_t) \right) \right] \quad (10)$$

where $sg(\cdot)$ is the stop-gradient operator and CE is the cross-entropy loss function. Reward prediction is formulated as a discrete regression problem in log-transformed space. Furthermore, $E_{\phi}$ and $F_{\phi}$ use SimNorm activation (Eq. 4) in their output layers. All trainable models are fully-connected MLPs with LayerNorm (Ba et al., 2016) and Mish activation (Misra, 2019). The complete algorithm is shown in Algorithm 1. Further implementation details can be found in Appendix C.

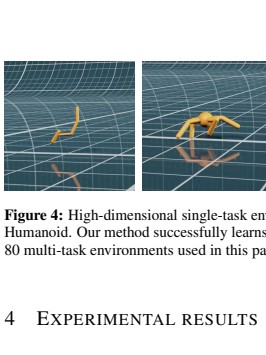 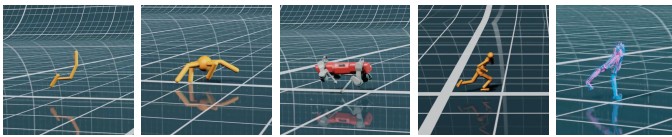

**Figure 4:** High-dimensional single-task environments (left to right): Hopper, Ant, Anymal, Humanoid and SNU Humanoid. Our method successfully learns tasks with up to $m = 152$ continuous action dimensions. Additional 80 multi-task environments used in this paper are listed in Appendix E

## 4 EXPERIMENTAL RESULTS

### 4.1 CONTACT-RICH SINGLE TASKS

We first assess our proposed method on complex continuous control tasks with up to $\mathcal{A} = \mathbb{R}^{152}$ using the differentiable simulator dflex (Xu et al., 2022). Hopper, Ant, Anymal, Humanoid and muscle-actuated (SNU) Humanoid (Figure 4) are tasked to maximize forward velocity. We compare against SHAC (Xu et al., 2022), a method with a similar actor-critic architecture as PWM but uses ground-truth dynamics and rewards from the simulation, instead of learning them. This allows us to understand whether world models induce better landscapes for policy learning. Furthermore, we compare against TD-MPC2 which uses the same world model but learns a policy in a model-free fashion and actively plans at inference time. This comparison allows us to understand whether first-order gradients can learn better policies. We additionally include prominent model-free baselines PPO (Schulman et al., 2017) and SAC (Haarnoja et al., 2018).

We conduct this experiment across 5 tasks with 5 seeds each where PWM and TD-MPC2 use the same pre-trained world models and are left to learn a policy and finetune their world models online. This is done to enable fair comparison to SHAC which directly has access to the simulation model and does not require any training. The results in Figure 5 reveal that (1) PWM is able to learn policies with higher reward than SHAC asymptotically, indicating that regularized world models induce smooth optimization landscapes than the true (discontinuous) dynamics. Furthermore (2) our method is able to learn policies with higher rewards than TD-MPC2 without the need for online planning and with the same compute time budget. However, PWM does not scale well to the highest dimensional task. More experiment details and results are included in Appendix D.

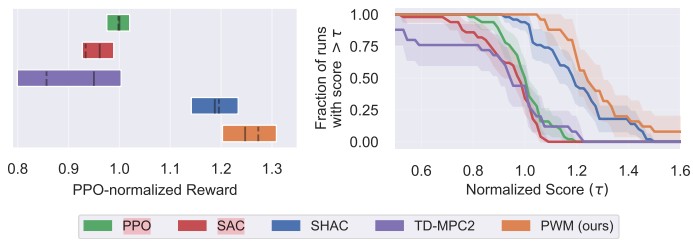

**Figure 5: Aggregate results from high-dimensional locomotion tasks** where each agent is trained to solve just that task (i.e. specialist). The **left** figure summarizes rewards achieved at the end of training using 50% IQM for the solid lines and 95% CI as suggested by (Agarwal et al., 2021), as well as mean for the dashed lines. We see that PWM achieves higher rewards than our main baselines TD-MPC2 and SHAC. The **right** figure shows score distributions across all tasks which lets us understand the performance variability of each approach. PWM exhibits a similar curve to SHAC but different than TD-MPC2, due to the policy learning approach.

---