# OpenReview forum: "PWM: Policy Learning with Multi-Task World Models"
_ICLR.cc/2025/Conference — ICLR 2025 Poster_

### Official Review · Reviewer_KWfz · 2024-10-22

**Soundness:** 3
**Presentation:** 2
**Contribution:** 2
**Rating:** 6
**Confidence:** 5

**Summary:**

This paper introduces PWM, proposing to use first-order gradient optimization for policy learning on top of offline pretrained world models. This method differs from zero-order optimization (MPPI) used in its base method TD-MPC2. Two didactic examples are presented to demonstrate that a well-regularized world model may be more useful for policy learning compared to real dynamics, confirming the well-known objective mismatch problem in the community. Main experimetns are conducted for single-task and multi-task learning, outperforming model-free baselines, zero-order method TD-MPC2, and method using real dynamics.

**Strengths:**

1. Didactic examples are informative.
2. Experiment results: sample efficiency to learn a task within 10 minutes seems promising.
3. The paper is well-written and easy to understand.

**Weaknesses:**

The major concerns are novelty and clarity of contributions. This paper does not well highlight its own contribution to the existing literature.

- The paper's contribution is conveyed (at least partially) on top of the fact that "world models methods often rely on inefficient gradient-free optimization methods for policy extraction." This holds for the TD-MPC series of work but is not valid for another series of popular methods, Dreamer. Actually, since DreamerV1, the policy learning process is done by first-order gradient optimization using the world model as a differentiable dynamics. The formulation of actor-critic learning in Equ (6-9) is almost the same as in the Dreamer series.
- Based on the above discussion, I believe the contribution of this work is additions to the following two problems: 1. objective mismatch in MBRL; 2. a multi-task RL with world models. However, contributions to this two aspect are also not groundbreaking. For the former, this problem has been well-known since Lambert et al. (2020), and for the latter, MBRL is also well-known for being beneficial for multi-tasking.

**Questions:**

Besides, there are several minor questions.

1. For the single-task setting, what's the difference between PWM and Dreamer (beyond implementation details)? Can PWM compare to DreamerV3 in the setting of Section 4.1?
2. In Line 298, do you mean world models are also finetuned with online interaction data? How about the multi-task setting? It seems the multi-task setting is fully offline.
3. In Figure 6 (left), why PWM significantly underperforms TD-MPC2?
4. In Line 357, does the baseline TD-MPC2 use the same hyperparameter as $H=16$?
5. Can the author explain why PWM benefits from smaller batch sizes?
6. Typo: Line 036 prpose => propose

Overall, I find this paper interesting and informative. If the authors address my concerns well, I am willing to raise my rating.

---

> ### Author Response · Authors · 2024-11-21
>
> We thank the reviewer for their valuable feedback. We address your comments in the following.
>
> > Q1: For the single-task setting, what's the difference between PWM and Dreamer (beyond implementation details)? Can PWM compare to DreamerV3 in the setting of Section 4.1?
>
> The focus of Section 4.1 is to “understand whether world models induce better landscapes for policy learning” than the true differentiable simulation, not to identify the best algorithm for these tasks.
> As such, we did not consider DreamerV3 [5] as a baseline but **will include results from it by the end of the rebuttal period**.
>
> The differences between PWM and DreamerV3 can be summarized as follows:
> 1. **World model:** PWM uses the TD-MPC2 world model, which is deterministic, learns by latent consistency loss, and is more aimed towards continuous control tasks. In contrast, DreamerV3 employs a stochastic, recurrent world model optimized via reconstruction loss, targeting both continuous control and game environments.
> 2. **Policy:** DreamerV3 trains its actor using zeroth-order optimization (Policy Gradients Theorem) and a stochastic critic with first-order optimization. PWM trains its actor with first-order optimization and its deterministic critic using zeroth-order (model-free) methods. Both use TD(\lambda) for the critic, but their actor objectives differ due to the optimization approaches. PWM is simpler than DreamerV3 and is more akin to the original Dreamer [4] with the major differences being the actor loss function and the fact that Dreamer still uses first-order optimization for the critic.
> 3. **Training Process:** DreamerV3 jointly trains its policy and world model from a replay buffer. In Section 4.1, PWM trains its world model from a buffer but learns its policy from observed data with gradients provided by the world model. This is done for fair comparison to SHAC and to answer the question whether the world model provides better dynamics gradients than the simulation.
>
> Given that (a) the aim of PWM is efficient multi-task learning and (b) the purpose of Section 4.1 is to show that world models can provide better gradients than ground truth dynamics, do you think the paper would benefit from a more thorough comparison to Dreamer?
>
> > Q2: In Line 298, do you mean world models are also finetuned with online interaction data? How about the multi-task setting? It seems the multi-task setting is fully offline.
>
> Yes, in Section 4.1, the world model is pre-trained on offline data, and then both the policy and world model are trained when interacting with the environment. “This is done to enable a fair comparison to SHAC, which directly has access to the simulation model and does not require any training”. Section 4.2 is fully offline RL.
>
> > Q3: In Figure 6 (left), why PWM significantly underperforms TD-MPC2?
>
> Thank you for highlighting them. We had used an incorrect figure, an issue we have rectified in the updated version. We would also like to point out that the correct results were still shown in Figures 1 and 14.
>
> > Q4: In Line 357, does the baseline TD-MPC2 use the same hyperparameter as H=16?
>
> No, TD-MPC2 uses the same hyperparameters as in the original work - $H=3$ and $\gamma=0.5$ [3].
>
> > Q5: Can the author explain why PWM benefits from smaller batch sizes?
>
> We currently lack a definitive explanation for this phenomenon. Suh et al. [1] suggest that an excess of stiff gradients from different samples may result in noisy gradients and inefficient optimization. Xu et al. [2] hypothesize that FoG methods are negatively impacted by the divergence of parallel environments across different state spaces, unlike ZoG methods like PPO, which generate a single gradient per state and are more robust. We recognize the importance of this question and believe it warrants further investigation.

---

> ### Author Response · Authors · 2024-11-21
>
> > The paper's contribution is conveyed (at least partially) on top of the fact that "world models methods often rely on inefficient gradient-free optimization methods for policy extraction." This holds for the TD-MPC series of work but is not valid for another series of popular methods, Dreamer. Actually, since DreamerV1, the policy learning process is done by first-order gradient optimization using the world model as a differentiable dynamics. The formulation of actor-critic learning in Equ (6-9) is almost the same as in the Dreamer series.
>
> Our primary contribution lies in understanding the properties world models need for efficient first-order optimization. To our knowledge, we are the first to study and identify the correlation between world model smoothness (via regularization) and policy performance. While the Dreamer series has indeed focused on first-order optimization, DreamerV2 and V3 have transitioned to zeroth-order optimization, with DreamerV3 relying entirely on the Policy Gradients Theorem.
>
> > Based on the above discussion, I believe the contribution of this work is additions to the following two problems: 1. objective mismatch in MBRL; 2. a multi-task RL with world models. However, contributions to this two aspect are also not groundbreaking. For the former, this problem has been well-known since Lambert et al. (2020), and for the latter, MBRL is also well-known for being beneficial for multi-tasking.
>
> We acknowledge Lambert et al. [6] as a significant influence on our work. While they highlight the weak correlation between world model accuracy and policy performance, our focus is on identifying the world model properties needed for efficient FoG optimization. We demonstrate that smoother, better-regularized world models significantly enhance policy performance. Coincidentally, this results in an inverse correlation between model accuracy and policy performance. Furthermore, we show that these regularized world models provide better policy gradients than the ground truth simulation. This makes our work distinctly novel from [6] and we are the first to extend small-scale toy examples to large-scale benchmarks and complex, high-dimensional tasks.
>
> Regarding multi-task MBRL, while its benefits have been shown recently [3], to our knowledge, we are the first to propose a framework for offline training a world model on multiple tasks and efficiently extracting performant per-task policies. Compared to existing approaches, PWM shows superior performance and offers a clear pathway to scale multi-task RL.
>
> [1] Suh et al. (2022), Do Differentiable Simulators Give Better Policy Gradients?
>
> [2] Xu et al. (2022), Accelerated Policy Learning with Parallel Differentiable Simulation
>
> [3] Hansen et al. (2024), TD-MPC2: Scalable, Robust World Models for Continuous Control
>
> [4] Hafner et al. (2020), Dream to Control: Learning Behaviors by Latent Imagination
>
> [5] Hafner et al. (2023), Mastering Diverse Domains through World Models
>
> [6] Lambert et al. (2020), Objective Mismatch in Model-based Reinforcement Learning

---

> ### Author Response · Authors · 2024-11-24
>
> Thank you all for your thoughtful reviews and feedback so far. As the rebuttal period is coming to a close, we would like to encourage further discussion or clarification on any remaining points. We are happy to address any concerns to ensure all perspectives are thoroughly considered.

---

> > ### Comment · Reviewer_KWfz · 2024-11-25
> >
> > I appreciate the detailed response provided by the author. Some of my concerns are addressed perfectly, but I still need further clarification on the following points:
> >
> > - Q1: I apologize for my outdated misunderstanding of DreamerV3. In the v1 of DreamerV3 arxiv paper, it claims, "We follow DreamerV2 in estimating the gradient of the first term by stochastic backpropagation for continuous actions and by reinforce for discrete actions"; However, in the v2 of arxiv paper, it turns to "We use the Reinforce estimator for both discrete and continuous actions". It seems DreamerV3 provides an opposite conclusion that reinforce policy gradient is preferred, in contrast to this paper that FoG is more efficient. Due to this debate, I believe this paper should be more conservative in claiming the discovery of FoG efficiency as its contribution since this needs more comprehensive experiments.
> > - What's more, since there are three separate contributions of this paper and the author emphasizes many times in the rebuttal that "correlation between world model smoothness (via regularization) and policy performance" is the highlighted contribution. The title of this paper should be revised to reflect the contribution better.
> > - Q4: Why do TD-MPC2 and PWM use different hyperparamters? Is the comparison fair? Does TD-MPC2 hurt from a longer horizon?
> >
> > I currently update my score to 5.

---

> ### Author Response · Authors · 2024-11-25
>
> We would like to thank the reviewer for engaging in the discussion and the overall positive response.
>
> > Q1: I apologize for my outdated misunderstanding of DreamerV3. In the v1 of DreamerV3 arxiv paper, it claims, "We follow DreamerV2 in estimating the gradient of the first term by stochastic backpropagation for continuous actions and by reinforce for discrete actions"; However, in the v2 of arxiv paper, it turns to "We use the Reinforce estimator for both discrete and continuous actions". It seems DreamerV3 provides an opposite conclusion that reinforce policy gradient is preferred, in contrast to this paper that FoG is more efficient. Due to this debate, I believe this paper should be more conservative in claiming the discovery of FoG efficiency as its contribution since this needs more comprehensive experiments.
>
> Thank you for bringing this to our attention—we were also unaware of this update in DreamerV3. However, extrapolating the conclusion that REINFORCE policy gradient is preferred may be premature. DreamerV3 addresses both continuous and discrete tasks, with a notable focus on discrete environments like Minecraft, where FoGs are not defined due to the lack of continuity in the state and action spaces. In such cases, the authors likely opted only for REINFORCE (ZoG) for algorithmic simplicity, albeit at the cost of performance in continuous control tasks.
>
> In contrast, PWM is explicitly designed for continuous control, where gradients are defined across most of the state and action space. Theoretically, FoGs are known to offer lower gradient variance [1], leading to superior sample efficiency and asymptotic performance. This has been  demonstrated empirically in the comparison between PPO (ZoG) and SHAC (FoG) [2]. Both algorithms are largely identical, and only differ in their actor optimization strategy where SHAC consistently outperforms PPO in asymptotic performance due to its FoG-based optimization (in differentiable simulation).
>
> PWM builds on these insights and shows that the benefits of FoG can also be extrapolated to learned world models. While we do think that 80 (multi) +5 (single) tasks is quite comprehensive, we are open to further **specific** suggestions on how to strengthen our results.
>
> > What's more, since there are three separate contributions of this paper and the author emphasizes many times in the rebuttal that "correlation between world model smoothness (via regularization) and policy performance" is the highlighted contribution. The title of this paper should be revised to reflect the contribution better.
>
> We understand your perspective and agree that there is merit to this suggestion. However, we believe the current title appropriately reflects the broader contributions of our work, including unlocking a new approach to scaling multi-task RL and achieving state-of-the-art results. We are excited to share these contributions with the scientific community to build upon.
>
> > Q4: Why do TD-MPC2 and PWM use different hyperparamters? Is the comparison fair? Does TD-MPC2 hurt from a longer horizon?
>
> We used the same hyperparameters for TD-MPC2 as Hansen et al. [3]. Limited experiments showed that while TD-MPC2 performs well asymptotically with $H=16$, it exhibits poorer sample efficiency. For PWM, $H$ has a greater influence, as it controls the extent of first-order gradient (FoG) optimization for the actor. A higher $H$ increases reliance on world model gradients, improving performance, while a lower $H$ relies more on the model-free critic. However, excessively high $H$ can increase gradient variance, as noted by Xu et al. [2]. To the best of our knowledge the different hyperparameters bring out the best of each algorithm, making for a fairer comparison.
>
> [1] Mohamed et al (2020), Monte Carlo Gradient Estimation in Machine Learning
>
> [2] Xu et al. (2022), Accelerated Policy Learning with Parallel Differentiable Simulation
>
> [3] Hansen et al. (2024), TD-MPC2: Scalable, Robust World Models for Continuous Control

---

> > ### Comment · Reviewer_KWfz · 2024-11-26
> >
> > Thanks for your extended response. My concerns have been well addressed, although I still think the scope of this work is not perfectly captured in its title. I have decided to update my score to a 6. This is the final score, and I cannot support it more.

---

### Official Review · Reviewer_d6vp · 2024-10-29

**Soundness:** 3
**Presentation:** 2
**Contribution:** 3
**Rating:** 6
**Confidence:** 4

**Summary:**

This paper introduces Policy learning with multi-task World Models (PWM) as a new model-based reinforcement learning algorithm. The main idea is to employ a pre-trained regularized and differentiable world model to simulate the environmental dynamics and then adopt first-order gradient methods to obtain the optimal policy efficiently. The author conducted experiments on both contact-rich single-task and large multi-task settings, showing that PWM had a better performance in learning high-reward policies and drastically improved the training time compared to previous methods. The author also provides additional examples and ablation studies to show the correlation between the accuracy of world models and the final performance, revealing that smoothness instead of accuracy is the key factor for higher rewards in gradient optimization.

**Strengths:**

1. The key idea of the paper is well delivered using pedagogical examples and is easy to follow

2. The results are highly reproducible with detailed experimental settings and code included, which also benefits the community for further study.

3. The experiments conducted in this paper are quite comprehensive and solid.

**Weaknesses:**

1. Typos and incorrect format
- Incorrect citing format in chapter 4.2: when the authors or the publication are included in the sentence, the citation should not be in parenthesis, please use \citet{} instead.
- Typos in Equation (2): it should be $\gamma^{h-t+1}$ instead of $\gamma^h$
- Typos in Algorithm 1: "$\nabla$" is omitted in all equations regarding learnable parameters update.
- Typos in Figure 10: missing ")" in the caption.

2. Lack of novelty
- The key idea of policy optimization via the pathwise derivative of a learned world model has already been proposed in MAAC[1], which is not included in the related works. The author should show the main differences between MAAC and the proposed PWM via analysis or experiments.

3. A total training time cost comparison between PWM and other algorithms is not revealed in the paper.

[1] Clavera, Ignasi, Yao Fu, and Pieter Abbeel. "Model-Augmented Actor-Critic: Backpropagating through Paths." International Conference on Learning Representations (2020).

**Questions:**

1. Since there are no interactions between the target policy and the environment, could this method be considered as an offline policy learning method? If so, how would this method solve the distributional shift problem? For example, will the world overestimate the value of some state-action pairs not included in the dataset and result in poor performance?
2. As mentioned in weakness 2, what will you consider to be the major differences between your method and MAAC?
3. I would like to know the comparison of the total training time (including world model training) of the PWM with other model-free methods like PPO and SAC.
4. How is "high-dimensional" defined in your work? For example, in Figure 4, "Hopper" is also considered a high-dimensional task. But according to the document (https://gymnasium.farama.org/environments/mujoco/hopper/), it only has an 11-dim observation space and a 3-dim action space.
5. According to Table 3 and Table 4, the PWM method shows much larger variances than model-free methods. What's the main reason for this and is it able to avoid the problem?
6. Some curves in Figure 13 are incomplete. What's the reason for that?

I am happy to increase my score if the authors justify these questions.

---

> ### Author Response · Authors · 2024-11-21
>
> We thank the reviewer for their valuable feedback. We address your comments in the following.
>
> > Typos and incorrect format
>
> Thank you for highlighting these. We have addressed them in the updated version.
>
> > Q1: Since there are no interactions between the target policy and the environment, could this method be considered as an offline policy learning method? If so, how would this method solve the distributional shift problem? For example, will the world overestimate the value of some state-action pairs not included in the dataset and result in poor performance?
>
> Yes, our approach can be considered within the domain of offline RL, but focuses on scaling multi-tasking. PWM does not directly address the distributional shift issue. Similar to findings in behavior cloning [1], we anticipate that training multi-task world models indirectly addresses the distributional shift issue as we are able to train over more data in comparison to a single-task setting. While this is an important direction for future work, it lies outside the scope of this paper.
>
> > Q2: As mentioned in weakness 2, what will you consider to be the major differences between your method and MAAC?
>
> Thank you for pointing this out. Many MBRL algorithms exist, making exhaustive comparisons challenging. MAAC shares similarities with PWM as both use first-order optimization over short horizons; one slight difference is that MAAC has an entropy bonus.
>
> PWM: $ \mathcal{L}_\pi(\theta) := \mathbb{E} \bigg[ \sum\_{h=1}^{H-1} \gamma^h r(z\_h, a\_h) + \gamma^H V (z\_H) \bigg]  $
>
> MAAC: $ \mathcal{L}_\pi(\theta) := \mathbb{E} \bigg[ \sum\_{h=1}^{H-1} \gamma^h r(z\_h, a\_h) + \gamma^H Q (z\_H, a\_H) \bigg]   + \beta \mathcal{H}(\pi\_\theta)$
>
> Additionally, MAAC employs a Q-critic trained with Model-based Value Expansion (MVE) [4], whereas PWM uses model-free TD($\lambda$).
>
> PWM: $\mathcal{L}_V(\psi) = \mathbb{E} \big[ \|\| V(z\_t) - \hat{V}(z\_t) \|\|\_2^2 \big]$ where $\hat{V}(z\_t)$ is obtained via TD($\lambda$), Eq 7-9.
>
> MAAC: $\mathcal{L}_Q(\psi) = \mathbb{E} \big[ \|\| Q(z\_t, a\_t) - (r(z\_t, a\_t) + \gamma \hat{Q}(z\_{t+1}, a\_{t+1}))  \|\|\_2^2\big] $ where $\hat{Q}$ is obtained via MVE (rolling out the model with gradients).
>
> Lastly, MAAC does not put much emphasis on the world model and how that is related to first-order optimization. Namely, they use an ensemble of Gaussian-parametarized dynamics models without any latent embedding. PWM trains its single deterministic world model by latent encoding and a latent consistency loss, making it simultaneously simpler and more scalable (for example, to other observation forms such as images).
>
>
> > Q3: I would like to know the comparison of the total training time (including world model training) of the PWM with other model-free methods like PPO and SAC.
>
> Below, we provide the total training times for SHAC, PPO, SAC, and TD-MPC2. All methods were trained until convergence. SHAC, PPO, and SAC follow the implementations and hyperparameter settings from Xu et al. [2], while TD-MPC2 is adapted for the vectorized setting of DFlex, as the original implementation by Hansen et al. [3] does not support this. Note that the training times are difficult to compare as different algorithms use different numbers of environments and need different amounts of steps to converge.
>
> However, we emphasize that the goal of PWM is not sample or time efficiency in online RL. Instead, Section 4.1 focuses on evaluating whether world models create better landscapes for policy learning compared to true differentiable simulation. Our primary interest lies in asymptotic performance (reward at convergence), as shown in Figure 5. Additionally, we note that the primary contributor to the higher training times of PWM and TD-MPC2 is the unoptimized vectorized interface to DFlex.
>
> | | Hopper | Ant | Anymal | Humanoid | SNU Humanoid
> | -- | -- | -- | -- | -- | -- |
> | PPO | 383 | 2,830 | 6,300 | 8,020 | 19,020 |
> | SAC | 1,230 | 10,700 | 3,190 | 7,380 | 7,380 |
> | TD-MPC2 |10,170 | 27,305 | 41,500 | 58,400 | 28,770 |
> | SHAC | 1,320 | 2,234 | 6,350 | 7,400 | 11,200 |
> | PWM | 7,740 |13,370 |27,477 |23,270 | 30,530 |
>
> **Total training time (seconds) until convergence**
>
> ||Hopper|Ant|Anymal|Humanoid|SNUHumanoid|
> |--|--|--|--|--|--|
> |PPO|56M|360M|196M|343M|520M|
> |SAC|4.1M|58M|6.3M|13.7M|13.7M|
> |TDMPC2|4.75M|30.2M|42.7M|60M|31M|
> |SHAC|13.7M|8M|8M|6.15M|10.24M|
> |PWM|13.7M|30.7M|28.4M|31M|31M|
>
> **Total steps**
>
> ||Hopper|Ant|Anymal|Humanoid|SNUHumanoid|
> |--|--|--|--|--|--|
> |PPO|6.84|7.86|32.14|23.38|36.58|
> |SAC|300|184.48|506.25|538.69|538.68|
> |TDMPC2|2141.05|904.14|971.90|973.33|928.06|
> |SHAC|96.35|279.25|793.75|1203.25|1093.75|
> |PWM|564.96|435.50|967.53|750.65|984.86|
>
> **Seconds / 1M steps**
>
> | | Hopper |  Ant | Anymal | Humanoid | SNU Humanoid |
> | -- | -- | -- | -- | -- | -- |
> |PPO|4096|4096|4096|4096|2048|
> |SAC|2048|128|64|64|256|
> |TD-MPC2|64|128|128|64|64|
> |SHAC|1024|128|128|64|64|
> |PWM|64|128|128|64|64|
>
> **Num envs**

---

> ### Author Response · Authors · 2024-11-21
>
> > Q4: How is "high-dimensional" defined in your work? For example, in Figure 4, "Hopper" is also considered a high-dimensional task. But according to the document (https://gymnasium.farama.org/environments/mujoco/hopper/), it only has an 11-dim observation space and a 3-dim action space.
>
> The definition of high-dimensional changes through time as we unlock new capabilities with newly proposed algorithms. In our work, we define it with the tasks that fall into Section 4.1 with up to 152 action dimensions. In comparison, the multi-task experiments in Section 4.2 are lower-dimensional, with the largest action space of 6.
>
> > Q5: According to Table 3 and Table 4, the PWM method shows much larger variances than model-free methods. What's the main reason for this and is it able to avoid the problem?
>
> Great question. We have two possible explanations: (1) PWM uses significantly fewer Monte Carlo samples for gradient estimation, leading to higher gradient and asymptotic performance variance. For example, PPO uses ~4k samples, while SHAC uses only 64 in most cases. While increasing sample numbers might reduce variance, the empirical results of Figure 7(b) suggests otherwise, highlighting an important direction for future research. (2) The more efficient FoG optimization in PWM may lead to the model becoming stuck in local minima, especially without sufficient exploration to escape, underscoring the impact of local minima in the world model.
>
> > Q6: Some curves in Figure 13 are incomplete. What's the reason for that?
>
> Thank you for highlighting this. The only truly incomplete curves are for TDMPC2 in the Ant and Humanoid experiments, which we have now updated. The SHAC curves are not incomplete in any of the experiments, as SHAC has already converged to its best performance according to [2] and our own experiments. In contrast, the other approaches need more samples to converge. We have clarified this in the updated version.
>
> [1] Open X-Embodiment: Robotic Learning Datasets and RT-X Models
>
> [2] Xu et al. (2022), Accelerated Policy Learning with Parallel Differentiable Simulation
>
> [3] Hansen et al. (2024), TD-MPC2:Scalable, Robust World Models for Continuous Control
>
> [4] Feinberg et al. (2018), Model-Based Value Estimation for Efficient Model-Free Reinforcement Learning

---

> ### Author Response · Authors · 2024-11-24
>
> Thank you all for your thoughtful reviews and feedback so far. As the rebuttal period is coming to a close, we would like to encourage further discussion or clarification on any remaining points. We are happy to address any concerns to ensure all perspectives are thoroughly considered.

---

> > ### Comment · Reviewer_d6vp · 2024-11-26
> >
> > I appreciate your clarifications and the additional results (although I believe the novelty should be explained and emphasized more when you submit the paper). I’m willing to increase my score.

---

### Official Review · Reviewer_WYHT · 2024-10-31

**Soundness:** 3
**Presentation:** 3
**Contribution:** 3
**Rating:** 6
**Confidence:** 3

**Summary:**

This paper learns a differentiable multi-task world model from offline data, and then learns a policy using first-order optimization through the world model. They find that a well-regularized world model (they use SimNorm activation to normalize the latent vector) that prioritizes smoothness over correctness of the transition predictions is more effective for learning policies. They explain this phenomenon with two pedagogical examples, related to discontinuous contact dynamics and chaotic dynamics. They demonstrate improvement over TD-MPC2, a leading MBRL agent published at the previous ICLR, and in some cases over SHAC, an approach that relies on ground truth world model differentiation.

**Strengths:**

- Good baseline comparisons with impressive results for PWM. The comparison to a method that uses ground truth dynamics gradients (from the differentiable simulator) is valuable and compelling.

- Good ablation experiments investigating sensitivity to the environment dynamics and the degree of world model regularization.

**Weaknesses:**

- A diagram of the model and policy training would be helpful (i.e. more detailed than Figure 1). Perhaps specifically illustrating what is different about this vs. TD-MPC2.

- I find Figure 2 confusing: What is the y-axis of the middle plot? What does a negative value mean? Is theta the angle (perhaps confusing because it is overloading the use of the theta symbol)? Additionally, I’m a little confused about how this example relates to a broader phenomenon of contact-induced discontinuities: the main problem here appears to be the long stretch of flat J, which causes the optimization to get stuck, as opposed to the fact that there is a non-differentiable discontinuity. Is it just a fluke of this toy scenario that the regularization helps? Why does SimNorm decrease the optimality gap? I’m a little ambivalent as to whether this example helps with understanding or just adds confusion, and on my initial read-throughs I didn’t find this “intuitive” example to be super intuitive, at least as-presented.

**Questions:**

1) Figure 6, MT80: is something wrong with this plot on the left? It look like PWM performs WAY worse, or am I misreading it? I am assuming this was a drafting error, but if it is not then it deserves more discussion.

2) The use of V in Eqn 4 is confusing, because I don’t think it has any relation to Eqn 2 (or does it?)

3) What are scenarios (if any) where this reduction in world model accuracy for the sake of smoothness could be problematic? I would appreciate a little more discussion. Perhaps for transfer between tasks that share dynamics but do not share any reward structure? Does the discrepancy between TDMPC2 and PWM on mw-stick-pull potentially point to this?

4) What might explain the reduced performance of PWM on the higher dimensional tasks, relative to SHAC (Figure 13)? Inaccuracies of the world model? If so, perhaps due to not having enough training data, or due to the regularization of the world model?

5) Where does the dataset come from? Is a source algorithm such as PPO used to gather the data? How much do these results depend on that? Do the datasets have examples of successful performance on each task? Does data from the task for which a policy is extracted need to have been included in the initial world model training?

7) Why can’t there be a comparison with DreamerV3 in a multi-task setting? In general, I am somewhat confused about how Dreamer, which I believe also uses first-order gradient optimization through a differentiable world model, relates to this work and would appreciate a little more discussion.

8) Is performance sensitive to the simnorm hyper parameters?


Minor:
- Some initial confusions (that were eventually cleared up, but hung around in my head for a while): what does '10 minutes' refer to? Of gameplay? Computation? On one GPU? Is new data acquired per task or it is reusing the pre-trained offline data for each specific task?
- I had to look at the specific references to understand that the first sentence of the intro was talking about foundation models. It is an interesting comparison, and I like the point, but could perhaps be made a little bit more obvious.
- In discussing large-data approaches to multi-task performance, I wonder how this relates to SIMA from the paper ‘Scaling Instructable Agents Across Many Simulated Worlds’?
- Typo in caption of Fig 1 (‘prpose’)
- Typo line 188 (‘optimality hap’)
- Typo line 377 (‘generalizble’)

---

> ### Author Response · Authors · 2024-11-21
>
> We thank the reviewer for their valuable feedback. We address your comments in the following.
>
> > I find Figure 2 confusing: What is the y-axis of the middle plot? What does a negative value mean? Is theta the angle (perhaps confusing because it is overloading the use of the theta symbol)?
>
> Figure 2b illustrates the problem landscape, i.e., the objective function $J(\theta)$ and its learned approximations, with $\theta$ being the starting angle (and the learnable parameter). While we’ve received positive feedback on the figures in Section 3, we welcome specific suggestions to further enhance their clarity and legibility.
>
> > Additionally, I’m a little confused about how this example relates to a broader phenomenon of contact-induced discontinuities: the main problem here appears to be the long stretch of flat J, which causes the optimization to get stuck, as opposed to the fact that there is a non-differentiable discontinuity.
>
> The ball-wall example we have aims to mimic sustained contact such as locomotion, where if using the most accurate contact models, we often get either 0 or very stiff (high) gradients [1,2,3].
>
> > Is it just a fluke of this toy scenario that the regularization helps?
>
> The ablation in Figure 8 and the extended results in Appendix F empirically demonstrate that additional regularization consistently aids in finding more optimal policies, particularly for complex and high-dimensional tasks.
>
> > Why does SimNorm decrease the optimality gap?
>
> SimNorm reduces the optimality gap by introducing additional regularization, which smooths discontinuities in the optimization objective and helps avoid local minima. Ultimately, this translates to a lower optimality gap at convergence. In the updated paper, we have revised Section 3.1 to clarify the connection between regularization and the optimality gap.
>
> > I’m a little ambivalent as to whether this example helps with understanding or just adds confusion, and on my initial read-throughs I didn’t find this “intuitive” example to be super intuitive, at least as-presented.
>
> The feedback on our pedagogical examples has been largely positive, especially with reviewers K87C and KWfz. We would appreciate if you have precise suggestions on how we can improve the readability of these examples
>
> > Q1: Figure 6, MT80: is something wrong with this plot on the left? It look like PWM performs WAY worse, or am I misreading it? I am assuming this was a drafting error, but if it is not then it deserves more discussion.
>
> Thank you for highlighting this. We had used an incorrect figure, an issue we have rectified in the updated version. We would also like to point out that the correct results were still shown in Figures 1 and 14.
>
> > Q2: The use of V in Eqn 4 is confusing, because I don’t think it has any relation to Eqn 2 (or does it?)
>
> Thank you for highlighting that. We acknowledge that it could lead to confusion and have changed it in Eq. 4.
>
> > Q3: What are scenarios (if any) where this reduction in world model accuracy for the sake of smoothness could be problematic? I would appreciate a little more discussion. Perhaps for transfer between tasks that share dynamics but do not share any reward structure? Does the discrepancy between TDMPC2 and PWM on mw-stick-pull potentially point to this?
>
> Great question! In our experiments, we have not observed scenarios where added regularization or smoothness has been problematic, likely due to the data-rich offline RL setting. However, in low-data regimes, this regularization could potentially introduce new local minima, pushing the RL agent into out-of-distribution state spaces. Investigating PWM under low-data, multi-task settings would be an interesting direction for future research.
>
> > Q4: What might explain the reduced performance of PWM on the higher dimensional tasks, relative to SHAC (Figure 13)? Inaccuracies of the world model? If so, perhaps due to not having enough training data, or due to the regularization of the world model?
>
> Figure 13 highlights that SHAC outperforms PWM in sample efficiency for high-dimensional tasks, but PWM often surpasses SHAC asymptotically. Although PWM starts with a pre-trained world model, it operates off-policy initially and relies on fine-tuning, which becomes more challenging in complex environments. In contrast, SHAC benefits from access to ground-truth dynamics from the start, explaining its superior sample efficiency. Notably, PWM’s ability to asymptotically outperform SHAC suggests that its world model provides more useful gradients than the simulation. Given more pre-training data, we are certain that PWM can outperform SHAC in sample efficiency.

---

> ### Author Response · Authors · 2024-11-21
>
> > Q5: Where does the dataset come from? Is a source algorithm such as PPO used to gather the data? How much do these results depend on that? Do the datasets have examples of successful performance on each task? Does data from the task for which a policy is extracted need to have been included in the initial world model training?
>
> We assume this question refers to Section 4.1 single-task experiments. From line 913, “We note that TDMPC2 and PWM use pre-trained world models on 20480 episodes of each task. The world models are trained for 100k gradient steps, and the same world models (specific to each task) are loaded into both approaches. The data consists of trajectories of varying policy quality generated with the SHAC algorithm. Trajectories include near-0 episode rewards as well as the highest reward achieved by SHAC. Note that we also run an early termination mechanism in these tasks, which is done to accelerate learning and iteration.”.
>
> > Does data from the task for which a policy is extracted need to have been included in the initial world model training?
>
> No, “This is done to enable fair comparison to SHAC which directly has access to the simulation model and does not require any training” from line 298.
>
> > Q6: Why can’t there be a comparison with DreamerV3 in a multi-task setting? In general, I am somewhat confused about how Dreamer, which I believe also uses first-order gradient optimization through a differentiable world model, relates to this work and would appreciate a little more discussion.
>
> Let us first highlight the differences between PWM and Dreamer; then we will expand on the multi-task setting.
> 1. **World model:** PWM uses the TD-MPC2 world model, which is deterministic, learns by latent consistency loss, and is more aimed towards continuous control tasks. In contrast, DreamerV3 employs a stochastic, recurrent world model optimized via reconstruction loss, targeting both continuous control and game environments.
> 2. **Policy Training:** DreamerV3 [5] trains its actor using zeroth-order optimization (Policy Gradients Theorem) and a stochastic critic with first-order optimization. PWM trains its actor with first-order optimization and its deterministic critic using zeroth-order (model-free) methods. Both use TD($\lambda$) for the critic, but their actor objectives differ due to the optimization approaches. PWM simpler than DreamerV3 and is more akin to the original Dreamer [4] with the major differences being the actor loss function and the fact that Dreamer still uses first-order optimization for the critic.
>
> To our knowledge, neither of the Dreamer algorithms have been applied to a multi-task setting. The most straightforward idea would be to condition all components (actor, critic, world model) on task embeddings and try to learn a multi-task policy similar to TD-MPC2. We attempted that in Appendix F.2, where instead of training a policy per task, we train a single multi-task policy across all tasks. As can be seen from Figure 18, PWM dissolves into a random policy. Given the similarities between this PWM ablation and Dreamer [4], we expect both of them to perform similarly.
>
> > Q7: Is performance sensitive to the simnorm hyper parameters?
>
> Not to our knowledge. Anything in the range of $P=[4, 36]$ has worked well.
>
> [1] Suh et al. (2022), Do Differentiable Simulators Give Better Policy Gradients?
>
> [2] Mason, M. T. (2001), Mechanics of Robot Manipulation
>
> [3] van der Schaft, A. and Schumacher, H. (200), An Introduction to Hybrid Dynamical Systems
>
> [4] Hafner et al. (2020), Dream to Control: Learning Behaviors by Latent Imagination
>
> [5] Hafner et al. (2023), Mastering Diverse Domains through World Models

---

> > ### Comment · Reviewer_WYHT · 2024-11-22
> > **Reply to comments**
> >
> > Thank you for your work discussing my questions.
> >
> > 1) I still think Fig 2 is confusing as is, though it’s unclear to me how exactly to improve it. Perhaps making more clear the definition of J(\theta) and r? And making clear why it is negative? And adding a ylabel to panel b? And using a different shape marker for the starting point of the optimization (since the parenthetical ‘marker x’ in the legend is not uniquely descriptive)? I may be overlooking where it is defined, but neither J nor r appear to be defined for this scenario in the main text or supplement.
> >
> > 2) Can you please add discussion (perhaps to the Limitations section) of scenarios where reduction in world model accuracy could be problematic?
> >
> > 3) Did you actually demonstrate that PWM asymptotically outperforms SHAC on the high dimensional tasks?  That doesn’t appear to me to be definitive. I am also still wondering whether this could be a scenario where a more accurate world mode is better.
> >
> > 4) Do you have evidence (perhaps I am overlooking it) that the data from the task for which a policy is extracted do NOT need to be included in the initial world model training? This could potentially be quite important for the generality of the world model.

---

> ### Author Response · Authors · 2024-11-24
>
> We would like to thank the reviewer for the prompt reply and ongoing discussion!
>
> > I still think Fig 2 is confusing as is, though it’s unclear to me how exactly to improve it. Perhaps making more clear the definition of J(\theta) and r? And making clear why it is negative? And adding a ylabel to panel b? And using a different shape marker for the starting point of the optimization (since the parenthetical ‘marker x’ in the legend is not uniquely descriptive)? I may be overlooking where it is defined, but neither J nor r appear to be defined for this scenario in the main text or supplement.
>
> Thank you for the clear feedback. First, we want to highlight that further details and exact definitions for Section 3.1 can be found in Appendix A, but we copy the main one of interest here.
>
> $$
> \begin{align}
>     J(\theta) = x_t = f(\theta) = \begin{cases}
>         x_0 + v \cos(\theta) t + \dfrac{1}{2} g t^2 & \text{if } y_{\text{contact}} > h \\
>         \quad \text{else} \quad w
>     \end{cases}
> \end{align}
> $$
>
> $$
> \begin{align*}
>     t_\text{contact} = \dfrac{-v cos(\theta) + \sqrt{v^2 \cos^2(\theta) + a w}}{a} &&
>     y_\text{contact} = y_0 + v\sin(\theta) t + \dfrac{1}{2} g t^2
> \end{align*}
> $$
>
> (please excuse the poor formatting from openreview)
>
> Since we are interested in maximizing $J(\theta)$ (or minimizing $-J(\theta)$), Figure 2b shows $-J(\theta)$ on the y axis to be consistent with prior work such as Suh et al. We have also added a * symbol to designate the start of the optimization process. Given this, do you find the following figure more easy to interpret? https://ibb.co/Db12gbR
>
> > Can you please add discussion (perhaps to the Limitations section) of scenarios where reduction in world model accuracy could be problematic?
>
> Yes, thank you for this good idea! We will add that for the camera-ready version.
>
> > Did you actually demonstrate that PWM asymptotically outperforms SHAC on the high dimensional tasks? That doesn’t appear to me to be definitive. I am also still wondering whether this could be a scenario where a more accurate world mode is better.
>
> We are firmly of the opinion that the results presented in Figure 5 are **strong empirical evidence** that PWM outperforms SHAC asymptotically. While our experiments only span only 5 environments, these are some of the most difficult and high-dimensional simulated problems RL has been evaluated on. We hope that these results will inspire further research on this topic.
>
> > Do you have evidence (perhaps I am overlooking it) that the data from the task for which a policy is extracted do NOT need to be included in the initial world model training? This could potentially be quite important for the generality of the world model.
>
> We value and understand your concern. We ran an ablation on the Ant task with and without the pre-trained world model. Both starting from an untrained policy. The results can be found here: https://ibb.co/MgDtQPz and follow the same format as our paper: 50% IQM with 95% CI, however, we only ran 3 seeds and only show 3 seeds for PWM with pre-training. The results show that PWM without pre-training is slower to reach a high reward but starts **converging towards the model with pre-training** (but has not converged due to the limited time).
>
> While these results are positive and show the potential capacity of PWM as an online algorithm, we also want to highlight that PWM is a multi-task offline RL algorithm. The focus of Section 4.1 is to “understand whether world models induce better landscapes for policy learning [...]” than the true differentiable simulation. We believe this question has already been answered in the paper without the need to explore pre-training.

---

> > ### Comment · Reviewer_WYHT · 2024-11-25
> >
> > Thank you for these updates, I do think they make things clearer, including the new labeling for 2b. And thank you for the new experiment.
> > Since '7' does not appear to an available score option to select, I would describe my score now as a 'high 6'. I do defer to the other reviewers, though, regarding the points that they raised.

---

### Official Review · Reviewer_K87C · 2024-11-01

**Soundness:** 3
**Presentation:** 3
**Contribution:** 2
**Rating:** 6
**Confidence:** 4

**Summary:**

This paper introduces a novel model-based reinforcement learning approach to pretrain a large multi-task world model on offline data with multiple tasks, followed by learning policies using first-order optimization for each task individually. Extensive experiments on DMC and Metaworld are conducted to showcase the strengths of the proposed method. While claiming to tackle multi-task challenges, this work tends to train a singular strategy for each isolated task.

**Strengths:**

1. This paper is clearly written and easy to follow.
2. This paper presents sufficient experimental results to demonstrate the validity of its proposed method.

**Weaknesses:**

1. My primary concern revolves around the novelty of the proposed approach. This study appears to amalgamate TD-MPC2 and SHAC methodologies. To elaborate, a multi-task world model incorporating a task embedding and SimNorm activation mirrors aspects of TD-MPC2 for representation learning. Subsequently, during the policy learning phase, policies are acquired via first-order optimization akin to SHAC. Consequently, the original contribution of this work may seem somewhat limited.
2. In Figure 13, PWM exhibits inferior performance compared to SHAC in high-dimensional tasks like Anymal, Humanoid, and SHU Humanoid. Does this observation suggest that in intricate environments, precise world models hold greater significance?
3. DreamerV3 is a powerful baseline model capable of handling various domains with a single configuration. It would be best to compare the proposed method with it in Figure 5.
4. In Figure 6, the left figure indicates that TM-MPC2 outperforms PWM in Metaworld, contradicting the statement "In both settings PWM achieves higher reward than TD-MPC2 without the need for online planning."
5. In Lines 348-349, the mean of 'm' and 'n' are not explicitly defined or mentioned.

**Questions:**

Please see the weaknesses section.

---

> ### Author Response · Authors · 2024-11-21
>
> We thank the reviewer for their valuable feedback. We address your comments in the following.
>
> > Q1. My primary concern revolves around the novelty of the proposed approach. This study appears to amalgamate TD-MPC2 and SHAC methodologies. To elaborate, a multi-task world model incorporating a task embedding and SimNorm activation mirrors aspects of TD-MPC2 for representation learning. Subsequently, during the policy learning phase, policies are acquired via first-order optimization akin to SHAC. Consequently, the original contribution of this work may seem somewhat limited.
>
> Our core contributions are twofold: (1) identifying the key world model properties necessary for efficient First-order Gradient (FoG) optimization and (2) introducing the multi-task PWM framework, which facilitates the efficient extraction of policies from a pre-trained multi-task world model. While PWM incorporates elements inspired by SHAC and TD-MPC2, to the best of our knowledge, we are the first to address these specific challenges and demonstrate contributions (1) and (2). We have revised the paper to clarify our overall contributions:
>
> 1. **Correlation Between World Model Smoothness and Policy Performance:** Building on prior work showing a weak correlation between model accuracy and policy performance, we demonstrate that smoother, better-regularized world models significantly enhance policy performance. Notably, this results in an inverse correlation between model accuracy and policy performance.
> 2. **Efficiency of First-Order Gradient (FoG) Optimization:** We show that combining FoG optimization with well-regularized world models enables more efficient policy learning compared to zeroth-order methods. Furthermore, policies learned from world models asymptotically outperform those trained with ground-truth simulation dynamics, emphasizing the importance of the tight relationship between FoG optimization and world model design. While FoG policy learning with world models has been explored, we focus more on the tight relationship between the two.
> 3. **Scalable Multi-Task Algorithm**: Instead of training a single multi-task policy model, we propose PWM, a framework where a multi-task world model is first pre-trained on offline data. Then per-task expert policies are extracted from it in <10 minutes per task using FoG optimization. This approach offers a clear and scalable alternative to existing methods focused on unified multi-task models.
>
> > Q2. In Figure 13, PWM exhibits inferior performance compared to SHAC in high-dimensional tasks like Anymal, Humanoid, and SHU Humanoid. Does this observation suggest that in intricate environments, precise world models hold greater significance?
>
> Figure 13 highlights that SHAC outperforms PWM in sample efficiency for high-dimensional tasks, but PWM often surpasses SHAC asymptotically. Although PWM starts with a pre-trained world model, it operates off-policy initially and relies on fine-tuning, which becomes more challenging in complex environments. In contrast, SHAC benefits from access to ground-truth dynamics from the start, explaining its superior sample efficiency. Notably, PWM’s ability to asymptotically outperform SHAC suggests that its world model provides more useful gradients than the simulation. Given more pre-training data, we surmise that PWM can outperform SHAC in sample efficiency.
>
> > Q3: DreamerV3 is a powerful baseline model capable of handling various domains with a single configuration. It would be best to compare the proposed method with it in Figure 5.
>
> The focus of Section 4.1 is to “understand whether world models induce better landscapes for policy learning” than the true differentiable simulation; rather than identifying the best algorithm that can solve these tasks. For this reason, we did not include DreamerV3 as a baseline but will add its results by the end of the rebuttal period.
>
> > Q4: In Figure 6, the left figure indicates that TD-MPC2 outperforms PWM in Metaworld, contradicting the statement "In both settings PWM achieves higher reward than TD-MPC2 without the need for online planning."
>
> Thank you for highlighting them. We had used an incorrect figure, an issue we have rectified in the updated version. We would also like to point out that the correct results were still shown in Figures 1 and 14.
>
> > Q5: In Lines 348-349, the mean of 'm' and 'n' are not explicitly defined or mentioned.
>
> $n$ and $m$ are defined on line 90 as the number of state and action dimensions, respectively.

---

> > ### Author Response · Authors · 2024-11-24
> >
> > Thank you all for your thoughtful reviews and feedback so far. As the rebuttal period is coming to a close, we would like to encourage further discussion or clarification on any remaining points. We are happy to address any concerns to ensure all perspectives are thoroughly considered.

---

> > > ### Comment · Reviewer_K87C · 2024-11-26
> > >
> > > Thank you for your response. It is better to highlight the section you revised in the revision. However, I still have some doubts regarding Q2. In Figure 13, both PWM and SHAC start training policies from 0 step, with PWM having undergone pre-training, which theoretically should lead to faster improvement. However, the results show that SHAC performs better. Your explanation is not convincing to me, so it would be beneficial to validate your claims through experiments.

---

> ### Author Response · Authors · 2024-11-26
>
> We would like to thank the reviewer for the prompt reply and ongoing discussion!
>
> > It is better to highlight the section you revised in the revision.
>
> We have uploaded the PDF difference highlighting all the revisions as supplementary material, as noted in our general response.
>
> > However, I still have some doubts regarding Q2. In Figure 13, both PWM and SHAC start training policies from 0 step, with PWM having undergone pre-training, which theoretically should lead to faster improvement. However, the results show that SHAC performs better.
>
> We appreciate your observation and would like to address it in more detail. To clarify, are you asserting that PWM, utilizing a learned world model, should universally outperform SHAC, which leverages ground-truth dynamics, in terms of sample efficiency? If this is the intended implication, we kindly ask for further clarification on your reasoning. Maybe it was not clear that in Section 4.1, we pre-train **only** the world model initially, after which both the world model and the policy (actor and critic) are trained online.
>
> > Your explanation is not convincing to me, so it would be beneficial to validate your claims through experiments.
>
> We are more than willing to conduct additional experiments to further validate our claims. To make the best use of these additional experiments, could you provide **specific** suggestions and explain how they would enhance the paper?
>
> From your response, we infer that you may wish to see whether PWM can match SHAC's sample efficiency with a better pre-trained world model (trained with more data). While we are open to exploring this direction, can you reason how such an experiment would contribute to the paper. As a reminder, the goal of Section 4.1 is to "understand whether world models induce better landscapes for policy learning". We do this in an online setting in order to compare to ground truth dynamics (i.e. SHAC) which can only be done in an online RL setting.

---

> > ### Comment · Reviewer_K87C · 2024-11-27
> >
> > Thank you once again for your response, and I apologize for my oversight. I will increase my score by 1.

---

### Author Response · Authors · 2024-11-21
**General response**

We thank the reviewers for their insightful feedback. We are glad that the reviewers found that our work:
1. Is clearly written and easy to follow (reveiwers K87C, d6vp, KWfz)
2. Uses pedagogical examples to build useful intuition (reveiwers d6vp, KWfz)
3. Presents extensive experimental results (reveiwers K87C, WYHT, d6vp)
4. Presents infightful ablations (reveiwer WYHT)
5. Offers an exciting avenue for scaling multi-task RL (reveiwer KWfz)

Despite the overall positive outlook, a common concern is the lack of novelty. As such, we would like to highlight that, to the best of our knowledge, our work is the first to show:
1. **Correlation Between World Model Smoothness and Policy Performance**: Building on prior work showing a weak correlation between model accuracy and policy performance [1], we demonstrate that smoother, better-regularized world models significantly enhance policy performance. Notably, this results in an inverse correlation between model accuracy and policy performance.
2. **Efficiency of First-Order Gradient (FoG) Optimization**: We show that combining FoG optimization with well-regularized world models enables more efficient policy learning compared to zeroth-order methods. Furthermore, policies learned from world models asymptotically outperform those trained with ground-truth simulation dynamics, emphasizing the importance of the tight relationship between FoG optimization and world model design. While FoG policy learning with world models has been explored [2,3,4], we focus more on the tight relationship between the two.
3. **Scalable Multi-Task Algorithm:** Instead of training a single multi-task policy model, we propose PWM, a framework where a multi-task world model is first pre-trained on offline data. Then per-task expert policies are extracted from it in <10 minutes per task using FoG optimization. This approach offers a clear and scalable alternative to existing methods focused on unified multi-task models.

We have addressed the reviewers’ concerns individually and updated our paper with the changes below. Lastly, we have included a version comparison PDF in the supplementary material.

**Summary of revisions:**
1. Rewritten our contributions to be more clear and concise.
2. We have updated Figure 6 to show the correct results.
3. Rewritten Section 3.1 to be more clear on the relationship between regularization and optimality gap as requested by reviewer WYHT.
4. Updated the results of TD-MPC2 in Section 4.1 for Ant and Humanoid tasks as requested by reviewer d6vp. The scores of TD-MPC2 are slightly improved, resulting in a more fair comparison between the approaches.

[1] Lambert et al. (2020), Objective Mismatch in Model-based Reinforcement Learning

[2] Hafner et al. (2020), Dream to Control: Learning Behaviors by Latent Imagination

[3] Iqbal et al. (2018), Actor-Attention-Critic for Multi-Agent Reinforcement Learning

[4] Amos et al. (2020), On the model-based stochastic value gradient for continuous reinforcement learning

---

### Author Response · Authors · 2024-11-28
**Preliminary DreamerV3 results**

We have updated the paper draft (and comparison in the supplimentary material) to incorporate preliminary results for DreamerV3 in Section 4.1, as requested by reviewers K87C and KWfz.

Currently, the algorithm has only converged on the Hopper and Ant tasks.
Please see the table below for IQM on the corresponding envs.
| Env |  DreamerV3 | PWM  |
| -------- | ------- | ------- |
| Hopper | 5321| **5680**  |
| Ant | 7009 | **9672** |

**These preliminary results further highlight the advantages of FoG optimization with smooth world models.**
However, we will reserve final conclusions until the full results are available.

---

### Meta-Review · Area_Chair_yz7L · 2024-12-19

**Metareview:**

This paper argues for the utility of well-regularized world-models for first-order optimization and propose Policy learning with multi-task World Models (PWM), consisting of pre-training a world-model on offline data and then extracting policies from it using first-order optimization. The authors present an empirical evaluation over a large number of tasks, comparing against related baselines.

The paper is well-written, explains concepts quite well (via some pedagogical examples), and provides strong empirical evaluations.

The main weakness of this work is its novelty. While the method is novel and results in strong empirical performance, none of its component parts (training world models, using first-order gradient optimization, and training on multi-tasks) is novel. The analysis in sections 3.1 and 3.2 is quite nice, but feel somewhat limited for one of the main claims of the paper ("Correlation Between World Model Smoothness and Policy Performance").

While their analysis does motivate their use in FoG optimization, the use of multiple tasks for training seems to come unnanounced. The fact that PWM works well in the multi-task setting does not answer _why_ the authors felt it necessary to add this. For instance, the pedagogical examples were all single-task settings, and this analysis was used to motivate the use of world-models for PWM.

In summary, this is an interesting idea that is well-presented, but I don't believe it is yet at its full potential. In particular, the correlation between world-model smoothness and policy performance bears digging into, as these insights will likely prove more useful to the general community than PWM itself. As such, I would personally recommend the authors extend this work with more analyses of that type so as to strengthen each of the three main claims.

However, given that the reviewers are unanimous in being supportive of acceptance, I will not override their decision.

**Additional Comments On Reviewer Discussion:**

There were a number of clarifications made by the authors on request by reviewers, as well as extra baselines considered (e.g. Dreamer). While the contributions were clarified, as mentioned above, the reviewers (and myself) still feel the overall novelty of and analyses are not as strong as they could be.

---

### Decision · Program_Chairs · 2025-01-22

Accept (Poster)